# Cytokinin transfer by a free-living mirid to *Nicotiana attenuata* recapitulates a strategy of endophytic insects

**Christoph Brütting[1†], Cristina Maria Crava[1†‡], Martin Schäfer[1§], Meredith C Schuman[1,2], Stefan Meldau[1#], Nora Adam[1,2¶], Ian T Baldwin[1]\***

[1]Department of Molecular Ecology, Max Planck Institute for Chemical Ecology, Jena, Germany; [2]German Centre for Integrative Biodiversity Research, Leipzig, Germany

**\*For correspondence:**
baldwin@ice.mpg.de

[†]These authors contributed equally to this work

**Present address:** [‡]Department of Biology and Biotechnology, University of Pavia, Pavia, Italy; [§]Institute for Evolution and Biodiversity, University of Münster, Münster, Germany; [#]Research & Development, Molecular Physiology, KWS SAAT SE, Einbeck, Germany; [¶]Research Group Sequestration and Detoxification in Insects, Max Planck Institute for Chemical Ecology, Jena, Germany

**Abstract** Endophytic insects provide the textbook examples of herbivores that manipulate their host plant's physiology, putatively altering source/sink relationships by transferring cytokinins (CK) to create 'green islands' that increase the nutritional value of infested tissues. However, unambiguous demonstrations of CK transfer are lacking. Here we show that feeding by the free-living herbivore *Tupiocoris notatus* on *Nicotiana attenuata* is characterized by stable nutrient levels, increased CK levels and alterations in CK-related transcript levels in attacked leaves, in striking similarity to endophytic insects. Using $^{15}$N-isotope labeling, we demonstrate that the CK $N^6$-isopentenyladenine (IP) is transferred from insects to plants via their oral secretions. In the field, *T. notatus* preferentially attacks leaves with transgenically increased CK levels; plants with abrogated CK-perception are less tolerant of *T. notatus* feeding damage. We infer that this free-living insect uses CKs to manipulate source/sink relationships to increase food quality and minimize the fitness consequences of its feeding.
DOI: https://doi.org/10.7554/eLife.36268.001

## Introduction

Insect herbivores are under constant pressure from their host plants: they must adapt to toxic or anti-digestive defense compounds whose levels often dramatically increase in response to insect feeding; and their food source has low nitrogen to carbon ratios and a dietary value which decreases as leaves mature and senesce. Some herbivorous insects have developed strategies to overcome the low nutritional contents of their host plants and have evolved specialized mechanisms to tolerate, or even co-opt toxic plant defense metabolites for their own uses, in an apparent evolutionary arms race (*Strong et al., 1984*; *Després et al., 2007*; *Heckel, 2014*).

Phytophagous insects can be categorized as either endophytic or free-living depending on the relationships that they establish with their host plant. This distinction is not binary and many transitional forms exist even within the same taxa. Consequently, the large differences in herbivorous lifestyles has selected for plant defense responses that counter different herbivory strategies (*Kessler and Baldwin, 2002*; *Schuman and Baldwin, 2016*). Free-living insects are mobile on their host plants, moving among plants, and frequently among different plant species. As a consequence of this mobility, they can freely choose tissues that are most nutritious or least defended, but the most nutritious tissues are often highly defended, resulting in a potential trade-off for herbivores (*Ohnmeiss and Baldwin, 2000*; *Brütting et al., 2017*). To avoid herbivore-induced defenses, free-living insects often move to other plant parts or even other host plants in response to defense activation, and the advantages of such movement are readily seen when induced defenses are abrogated (*Paschold et al., 2007*) or experimentally manipulated (*van Dam et al., 2000*). In contrast,

**eLife digest** Many insects use plants for food and for shelter. To protect themselves, plants often develop defense mechanisms that deter or debilitate their attackers, such as producing toxins or storing nutrients away from the attacked tissues.

But some insects manage to counter the plants' defense responses. Such species are often less mobile and spend a large part of their life in a restricted area of the plant, for example, inside plant tissues. Also known as 'endophytic' animals, these insects can even manipulate the signaling system in a plant, such as a class of plant hormones called cytokinins, which help plants to grow and to develop seeds and nutrient-storing fruits or young leaves.

Researchers have previously assumed that endophytic animals target cytokinins because they are restricted to living in certain areas within the plant, and – unlike 'free-living' insects – lack access to other, potentially more nutritious feeding sites. By modifying cytokinins, the location-bound insects could create their own 'nutrient pool'. Until now, it was unclear how insects transfer cytokinins to a plant and if this ability was restricted to endophytic insects.

To investigate this further, Brütting, Crava et al. studied the response of coyote tobacco plants infested with a free-living insect, the sap-sucking bug *Tupiocoris notatus*. The experiments revealed that the insects' bodies contained large quantities of a type of cytokinin, even when insects were raised on artificial diets.

Brütting, Crava et al. then developed a method to clearly distinguish cytokinins present in the insects from those produced by the plants to test whether *T. notatus* can transfer these plant hormones during feeding. The results showed that similar to endophytic insects, *T. notatus* injects cytokinins into the attacked leaves, presumably to create a stable nutritious environment.

Insects represent the largest and most diverse group of organisms on Earth, including many crop pests. Despite their detrimental impact on the environment and the health of the farmers, pesticides are used predominantly for pest control. A better understanding of how insects use cytokinins to increase the nutritional value of the leaves may help us to find ways to increase the crop's tolerance to insect attacks.

DOI: https://doi.org/10.7554/eLife.36268.002

endophytic insects develop more intimate relationships with their host plants as they are sedentary and spend a large portion of their life cycle within plant tissues. They have evolved strategies to overcome many of the plant defenses by hijacking plant metabolism and reprogramming plant physiology in their favor (*Giron et al., 2016*). Often the only viable plant defense is the 'scorched earth' response, whereby infested tissues are abscised from the plant (*Fernandes et al., 2008*).

To date, the best-studied examples of endophytic plant-manipulating species, featured in most textbooks of plant physiology, are the gall-forming insects and leaf-miners. Gall-forming organisms, which include not only several orders of insects but also mites, nematodes and microbes, promote abnormal plant growth by reprogramming the expression of plant genes, to create novel organs that provide favorable environments for the exploiter (*Stone and Schönrogge, 2003*; *Shorthouse et al., 2005*). Advantages for the gall-formers range from an improved nutritional value, with reduced defense levels, to protection from diseases, competitors, predators, parasitoids and unfavorable abiotic conditions (*Hartley, 1998*; *Stone and Schönrogge, 2003*; *Allison and Schultz, 2005*; *Harris et al., 2006*; *Saltzmann et al., 2008*; *Nabity et al., 2013*). Manipulations of leaf-mining larvae do not result in the formation of new macroscopic structures like galls but they are often revealed during senescence of host tissues, where 'green islands' appear around the active feeding sites (*Engelbrecht, 1968*; *Engelbrecht et al., 1969*; *Giron et al., 2007*; *Kaiser et al., 2010*). Such green islands maintain a high level of photosynthetic activity typical of non-senescent leaves, thus providing nutrition for the larvae which feed on them (*Behr et al., 2010*; *Body et al., 2013*; *Zhang et al., 2016*). In this way, green islands reflect a battle between plant and infesting insect during the nutrient recovery phase that precedes abscission. The host plant tries to recover nutrients from the senescent leaf, whereas the insect tries to maintain a nutritious environment so as to complete its development.

The most likely effectors used by insects to manipulate a plant's normal physiological response to wounding are phytohormones, since significant levels of some well-known wound-responsive phytohormones, including cytokinins (CKs), abscisic acid (ABA) and auxins, have been found in the body and salivary secretions of a number of gall-forming insects (*Mapes and Davies, 2001*; *Straka et al., 2010*; *Tooker and De Moraes, 2011*; *Yamaguchi et al., 2012*; *Tanaka et al., 2013*; *Takei et al., 2015*), as well as in the bodies and labial glands of leaf-mining larvae (*Engelbrecht et al., 1969*; *Body et al., 2013*). Amongst these phytohormones, CKs deserve additional discussion due to their role in the formation of green islands (*Engelbrecht, 1968*; *Engelbrecht, 1971*; *Engelbrecht et al., 1969*; *Giron et al., 2007*; *Kaiser et al., 2010*; *Body et al., 2013*; *Zhang et al., 2017*). CKs are adenine derivatives which play a key role in the regulation of plant growth and development (*Sakakibara, 2006*). They are known for their capacity to increase photosynthetic activity (*Jordi et al., 2000*), determine sink strength (*Mok and Mok, 2001*) and inhibit senescence (*Richmond and Lang, 1957*; *Gan and Amasino, 1995*; *Ori et al., 1999*). More recently, CKs have been shown to regulate herbivory-induced defense signaling (*Schäfer et al., 2015b*; *Schäfer et al., 2015c*; *Brütting et al., 2017*). The long history of investigating CKs in the formation of green islands dates back to the late 1960's, to reports of increased levels of CKs in affected tissues (*Engelbrecht, 1968*; *Engelbrecht et al., 1969*). In the last decade, studies on the leaf-mining larvae of *Phyllonorycter blancardella* identified CKs as the causative factors for the 'green island' phenomenon (*Giron et al., 2007*; *Kaiser et al., 2010*; *Body et al., 2013*; *Zhang et al., 2017*). These studies suggested that insects could be the source of phytohormones used to manipulate plant physiological responses. However, a clear demonstration of the ability of insects to transfer CKs to a host plant remains elusive.

To assess whether an insect actively transfers CKs to manipulate plant physiology, we studied the interactions between the well-established ecological model-plant *Nicotiana attenuata* and one of its most abundant specialist herbivores, *Tupiocoris notatus*. *N. attenuata* is a wild diploid tobacco species native to southwestern North America. *T. notatus* is a free-living, 3–4 mm mirid bug (Miridae, Heteroptera) specialized to tobacco species and a few other solanaceous plants including *Datura wrightii*. It is a piercing-sucking cell-content feeder that damages the surface of the leaves without removing foliar material. Its feeding behavior is in sharp contrast with the feeding behavior of a well-studied specialist herbivore of *N. attenuata*, the lepidopteran *Manduca sexta*, whose chewing larvae cause extensive tissue damage and a well characterized defense response (*Baldwin, 1998*; *Kessler and Baldwin, 2001*; *Kessler et al., 2004*; *Steppuhn et al., 2004*; *Zavala et al., 2004*; *Schuman et al., 2012*). When plants are attacked by *M. sexta*, specific insect-derived fatty acid-amino acid conjugates elicit a defense response regulated by a burst of jasmonic-acid (JA) (*Baldwin, 1998*; *Halitschke et al., 2001*; *Kessler et al., 2004*). This jasmonate burst triggers the accumulation of defense metabolites like nicotine, caffeoylputrescine, diterpene-glycosides and tripsyin-proteinase inhibitors. It has also strong effects on the regulation of primary metabolism (*Voelckel and Baldwin, 2004*): sugars, starch and total soluble proteins readily decrease in the attacked leaves (*Ullmann-Zeunert et al., 2013*; *Machado et al., 2015*), as does photosynthesis (*Meza-Canales et al., 2017*).

In contrast, infestation with *T. notatus* in nature, surprisingly, does not decrease plant fitness (*Kessler and Baldwin, 2004*), despite resulting in damage to large portions of photosynthetically active leaf area. Tissues around *T. notatus* feeding sites have increased rates of photosynthesis per chlorophyll content that may compensate for the damage caused by herbivore feeding, resulting from an active ingredient of the oral secretion of *T. notatus* which remains to be identified (*Halitschke et al., 2011*). We previously observed increased damage by *T. notatus* in tissues that were enriched in CKs through the transgenic manipulation of *N. attenuata* CK metabolism, using plants expressing a dexamethasone (DEX)-inducible construct driving transcription of the CK biosynthesis gene, isopentenyltransferase (IPT, i-ov*ipt*). Individual DEX-treated leaves of field-grown plants suffered more damage from *T. notatus* than did mock-treated leaves. This led to the hypothesis that increased CK levels promote better nutritional quality, which in turn increases *T. notatus* feeding damage (*Schäfer et al., 2013*).

Here, we report that *T. notatus* adults and nymphs contain high concentrations of two CKs. When confined to feeding on single *N. attenuata* leaves, concentrations of CKs increase in attacked leaves throughout the feeding period, with consequences for nutrient concentrations. Using $^{15}$N-labeled tracers, we demonstrate that *T. notatus* transfer CKs to the leaves on which they feed. Finally, we

analyzed how changes to CK metabolism in plants affected *T. notatus* feeding preferences. We conclude that CK-dependent manipulation of plant metabolism is not only a strategy used by gall-forming insects or leaf-miners, but also employed by this free-living insect, which directly transfers CKs at feeding sites to manipulate its host plant.

## Results

### *Tupiocoris notatus* feeding induces the JA pathway and associated defenses in *Nicotiana attenuata*

To characterize the defensive response of *N. attenuata* to mirid attack, we analyzed jasmonate hormones and defense metabolites that are known to be induced by *M. sexta,* as well as *T. notatus* feeding (*Kessler and Baldwin, 2004*). Continuous feeding by *T. notatus* (*Figure 1a*) causes visible damage to *N. attenuata* leaves (*Figure 1b*) and triggers defense responses in attacked leaves (*Figure 1c–j*). Three days of *T. notatus* feeding induced levels of the defense metabolites nicotine and caffeoylputrescine (CP), as well as trypsin proteinase inhibitor activity (TPI) (*Figure 1c–e*). *T. notatus* feeding also elevated the levels of jasmonic acid (JA), its precursor *cis*-(+)−12-oxophytodienoic acid (OPDA) and its bioactive isoleucine conjugate (JA-Ile) (*Figure 1f–h*). Interestingly, there was also a significant increase in salicylic acid (SA), but no influence on abscisic acid (ABA) (*Figure 1i,j*). JA and JA-Ile levels triggered by *T. notatus* feeding remained elevated for up to six days when mirids were confined to feed on a single leaf (*Figure 1—figure supplement 1a–c*). Their concentrations remained higher than controls even when *T. notatus* were free to move to other parts of the plant, although they steadily decreased over the six days (*Figure 1—figure supplement 1d–f*) These results demonstrate that *N. attenuata*'s response to *T. notatus* involves activation of JA signaling and downstream defense responses.

### *T. notatus* feeding does not negatively affect the nutritional quality of the attacked leaves

Feeding by *M. sexta* is detrimental to *N. attenuata* fitness. It causes reduction of photosynthesis in attacked leaves (*Halitschke et al., 2011*; *Meza-Canales et al., 2017*) and a decrease in sugar and total soluble protein (TSP) contents (*Ullmann-Zeunert et al., 2013*; *Machado et al., 2015*). In contrast, *T. notatus* feeding seems to increase photosynthetic activity in attacked leaves, when accounting for tissue damaged by the feeding (*Halitschke et al., 2011*). We measured the impact of continuous *T. notatus* feeding over several days on the nutritional quality of the attacked leaves. We analyzed TSPs, sugar and starch levels, as well as measuring photosynthetic rates and chlorophyll contents of leaves over a period of 144 hr.

Visibly heavily damaged leaves did not show significant decreases in nutrient levels when mirids were confined to feed on a single leaf with a small plastic cage (*Figure 2a*). TSP levels decreased with time in a clipcage but mirid feeding did not have a significant influence (*Figure 2b*). Furthermore, we did not observe any significant changes in starch, sucrose, glucose or fructose (*Figure 2c–f*). Although we did not observe changes in carbohydrate levels, photosynthesis was significantly reduced in attacked leaves (*Figure 2—figure supplement 1b*). In contrast, mirid feeding had no effect on chlorophyll contents (*Figure 2—figure supplement 1c*).

When entire plants were heavily infested (*Figure 2—figure supplement 2a*), changes in nutrient levels in the plant became apparent only for TSP levels, which decreased after mirid feeding (*Figure 2—figure supplement 2b*). Conversely, levels of starch, sucrose, glucose and fructose were not affected by mirid feeding (*Figure 2—figure supplement 2c–f*). Both chlorophyll contents and photosynthetic rates significantly decreased after *T. notatus* whole-plant attack (*Figure 2—figure supplement 3a–c*).

In summary, when only twenty mirids were allowed to feed on a single leaf the overall nutritional quality was not altered, although the feeding damage was visibly severe. In contrast, during a more extreme mirid infestation in which entire plants were severely attacked, TSP levels of attacked leaves decreased, but sugar and starch contents remained unchanged. However, no overall apparent increased photosynthetic activity was observed. An allocation of nutrients from unattacked to attacked tissue may explain the observation that even heavy *T. notatus* feeding only marginally

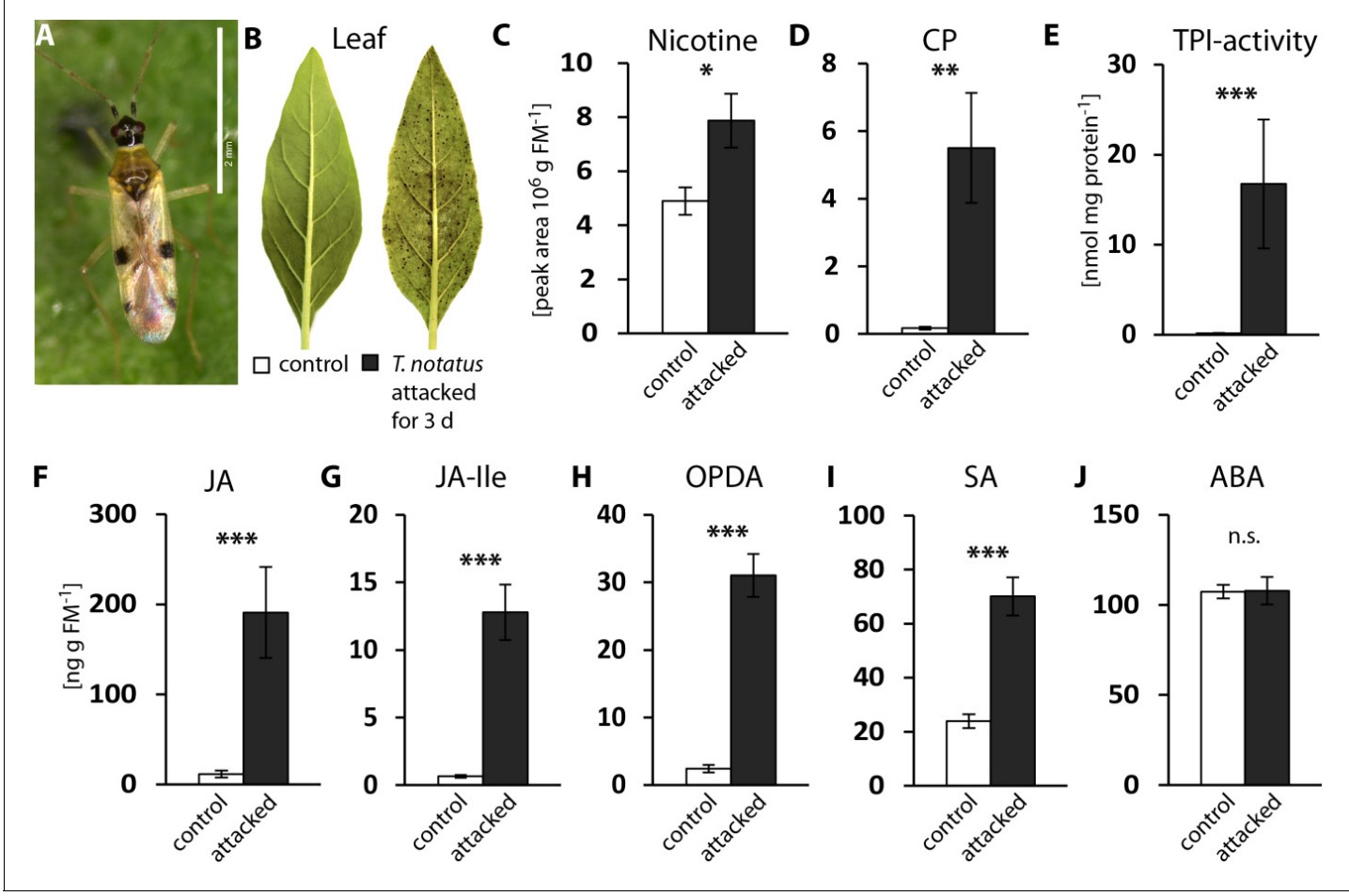

**Figure 1.** *Tupiocoris notatus* feeding induces JA-dependent defense responses in *Nicotiana attenuata*. (A) *T. notatus* adult. (B) Representative pictures of a control leaf of *N. attenuata* and a leaf after 3 d of continuous *T. notatus* feeding. (C–J) Defense metabolites and stress-related phytohormone levels induced by 3 d of *T. notatus* attack (filled columns) compared with control leaves (open) from unattacked plants: (C) nicotine, (D) caffeoylputrescine (CP), (E) trypsin proteinase inhibitor (TPI) activity, (F) jasmonic acid (JA), (G) jasmonic acid-isoleucine conjugate (JA-Ile), (H) *cis*-(+)−12-oxophytodienoic acid (OPDA), (I) salicylic acid (SA) and (J) abscisic acid (ABA). Wilcoxon-Mann-Whitney test was used to identify statistically significant differences between control and attacked leaves. (C) nicotine: N = 6, W = 5, p=0.022; (D) CP: N = 6, W = 0, p=0.001, (E) TPI: N = 7, W = 0, p<0.001, (F) JA: N = 7, W = 0, p<0.001, (G) JA-Ile: N = 7, W = 0, p<0.001; (H) OPDA: N = 7, W = 0, p<0.001; (I) SA: N = 7, W = 0, p<0.001: (J) ABA: N = 7, W = 20, p=0.620. *p<0.05, **p<0.01, ***p<0.001, n.s.: not significant. Error bars depict standard errors. FM: fresh mass. For raw data see Raw_data_FIGURE_1 (Dryad: *Brütting et al., 2018*).

DOI: https://doi.org/10.7554/eLife.36268.003

The following figure supplement is available for figure 1:

**Figure supplement 1.** *Tupiocoris notatus* feeding increases levels of JA and JA-Ile.

DOI: https://doi.org/10.7554/eLife.36268.004

influenced nutrient levels in attacked leaves. If this inference is correct, then mirid feeding likely influences the source/sink relationships of the host plant.

## *T. notatus* attack increases the levels of cytokinins and transcripts responsible for cytokinin degradation

Cytokinins (CKs) are known to regulate source/sink relationships and stabilize nutrient levels in tissues fed on by endophytic insects. Recently, we showed that these phytohormones also play a role in plant defense, since *M. sexta* herbivory, wounding, and JAs can increase the levels of *cZ*-type CKs in *N. attenuata* (*Schäfer et al., 2015c*; *Brütting et al., 2017*). As we did not see a strong decrease

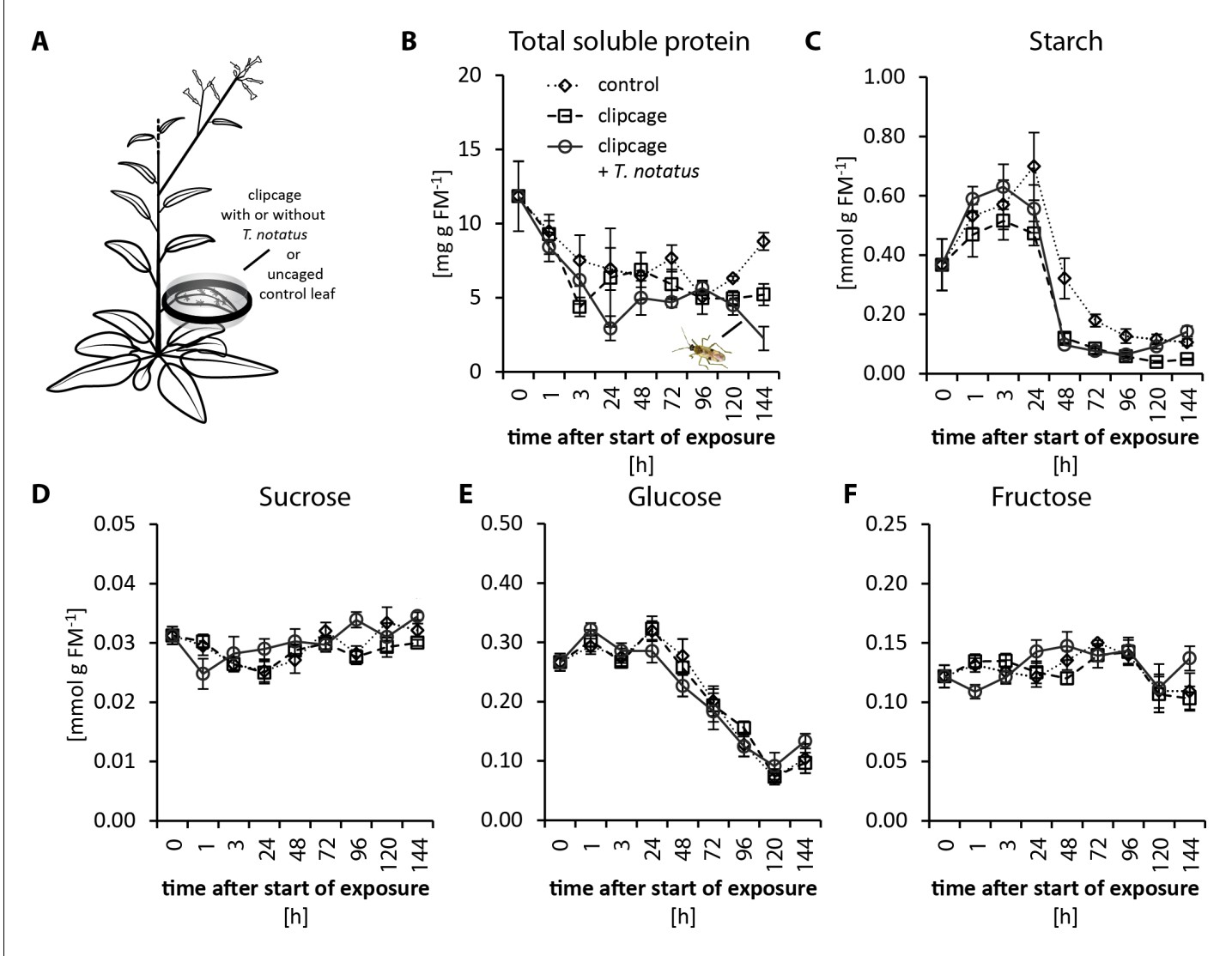

**Figure 2.** *Tupiocoris notatus* feeding on single leaves does not significantly change nutrient levels. (**A**) Experimental setup: On each plant we enclosed one leaf in a plastic clipcage with (clipcage +*T. notatus*; solid line) or without (clipcage, dashed line) 20 *T. notatus*. Additionally, we collected uncaged control leaves (control, dotted line). (**B**) Total soluble proteins (TSP), (**C**) starch, (**D**) sucrose, (**E**) glucose and (**F**) fructose were analyzed in a time-kinetic from 1 to 144 hr. Statistically significant differences were identified with ANCOVA with mirid as factor and time as continuous explanatory variable. (**B**) TSP: log transformed time $F_{1,51}$ = 14.317, p<0.001; mirid $F_{1,51}$ = 2.438, p=0.125; time*mirid $F_{1,51}$ = 0.479, p=0.492. (**C**) log transformed starch: time $F_{1,51}$ = 137.376, p<0.001; mirid $F_{1,51}$ = 3.749, p=0.058; time*mirid $F_{1,51}$ = 2.651, p=0.110. (**D**) sucrose: time $F_{1,51}$ = 13.847, p<0.001; mirid $F_{1,51}$ = 3.883, p=0.054; time*mirid $F_{1,51}$ = 5.894, p=0.019. (**E**) glucose: time $F_{1,51}$ = 173.06, p<0.001; mirid $F_{1,51}$ = 0.050, p=0.823; time*mirid $F_{1,51}$ = 0.107, p=0.745. (**F**) fructose: log transformed time $F_{1,51}$ = 0.505, p=0.480; mirid $F_{1,51}$ = 0.433, p=0.513; time*mirid $F_{1,51}$ = 5.798, p=0.020. Error bars depict standard errors (N ≥ 3). FM: fresh mass. For raw data see Raw_data_FIGURE_2 (Dryad: ***Brütting et al., 2018***).

DOI: https://doi.org/10.7554/eLife.36268.005

The following figure supplements are available for figure 2:

**Figure supplement 1.** *Tupiocoris notatus* feeding on single leaves decreases photosynthetic rates while not influencing chlorophyll contents.
DOI: https://doi.org/10.7554/eLife.36268.006

**Figure supplement 2.** *Tupiocoris notatus* feeding on whole plants only slightly alters nutrient levels in attacked leaves of *Nicotiana attenuata*, mainly decreasing protein contents.
DOI: https://doi.org/10.7554/eLife.36268.007

**Figure supplement 3.** *Tupiocoris notatus* feeding on whole plants decreases photosynthetic rates and chlorophyll contents in attacked leaves of *Nicotiana attenuata*.
DOI: https://doi.org/10.7554/eLife.36268.008

in nutrients after mirid feeding, it was especially interesting to investigate CK metabolism during *T. notatus* attack.

When entire plants were attacked, mirid feeding significantly increased the accumulation of *NaCKX5* transcripts, which code for a CK oxidase/dehydrogenase responsible for CK degradation (*Figure 3a*). Transcript levels of *NaZOG2*, which codes for a CK glucosyltransferase responsible for CK inactivation, as well as transcripts of *NaLOG4* (*Figure 3—figure supplement 1b*), which is involved in CK biosynthesis, also increased after mirid feeding. In contrast, transcript levels of the isopentenyltransferase *NaIPT5,* which catalyzes the rate-limiting step of CK biosynthesis, were reduced after mirid feeding (*Figure 3—figure supplement 1c*). *T. notatus* feeding did not change levels of the CK response regulator *NaRRA5* (*Figure 3—figure supplement 1d*).

The levels of the different types of CKs varied depending on time and whether mirids attacked single leaves or entire plants. When entire plants were infested with *T. notatus*, overall leaf CK

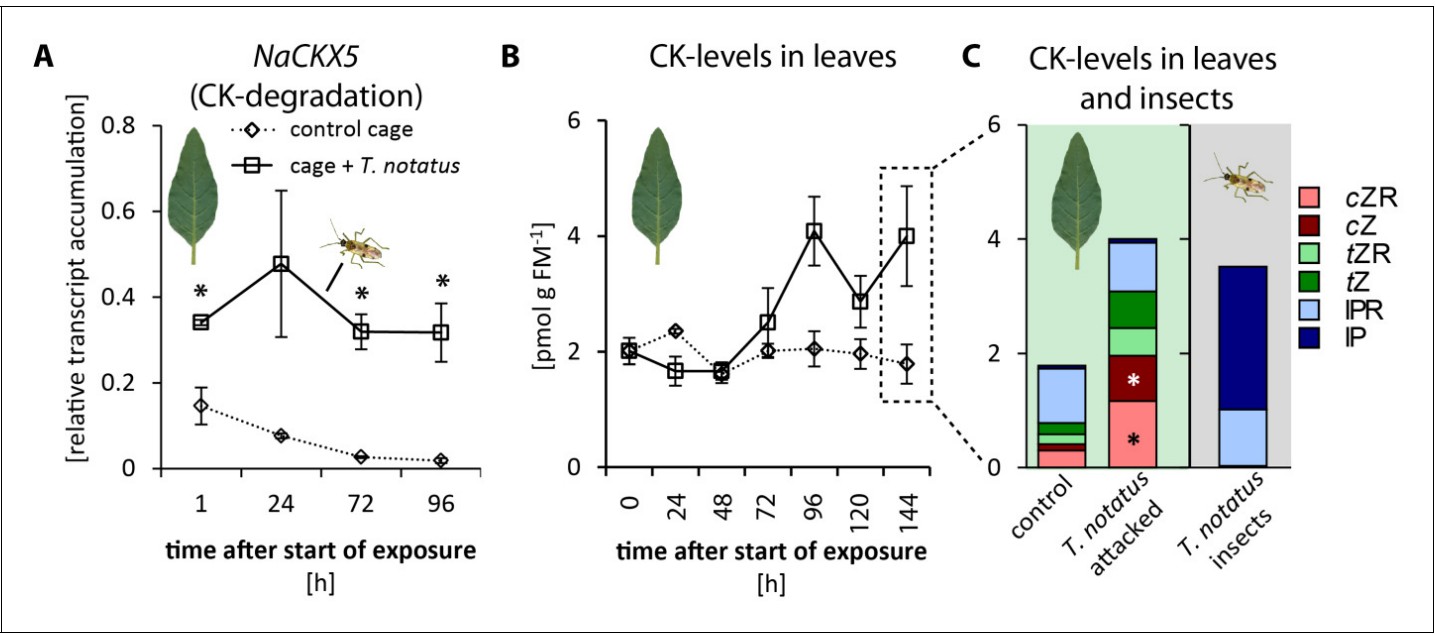

**Figure 3.** *Tupiocoris notatus* contain large amounts of CKs in their bodies and their feeding alters *Nicotiana attenuata's* cytokinin (CK) metabolism. (A) Transcript accumulations of *NaCKX5*: cytokinin oxidase/dehydrogenase 5 (which inactivates CKs by oxidation) and (B) CK levels in leaves: sum of *cis*-zeatin (*cZ*), *trans*-zeatin (*tZ*), $N^6$-isopentenyladenine (IP) and their ribosides (*cZR, tZR,* IPR) in leaves exposed to *T. notatus* feeding (cage +*T. notatus,* solid line) and control leaves (control cage, dotted line) at different times after herbivore exposure. (C) Single CK types in leaves after 144 hr of exposure to *T. notatus* and in the insect bodies. Two-way ANOVA on $\log_2$-transformed data followed by Welch t-test with Bonferroni corrections between control and *T. notatus* cage for each harvest time were used to analyze A) (mirid $F_{1,13}$ = 158.2, p<0.001; time $F_{3,13}$ = 12.52, p<0.001; time*mirid $F_{3,13}$ 7.015, p=0.005). ANCOVA with mirid as factor and time as continuous explanatory variable was used to analyze B) ($\log_2$ transformed data: time $F_{1,44}$ = 7.335, p=0.010; mirid $F_{1,44}$ = 8.609, p=0.005; time*mirid $F_{1,44}$ = 15.243 *p*<0.001). (C) was analyzed with Wilcoxon-Mann-Whitney test between control and attacked leaves for each CK type. • p<0.1, *p<0.05. Error bars depict standard errors (A): N ≥ 2; (B, C): N = 4. FM: fresh mass. For raw data see Raw_data_FIGURE_3 (Dryad: *Brütting et al., 2018*).

DOI: https://doi.org/10.7554/eLife.36268.009

The following figure supplements are available for figure 3:

**Figure supplement 1.** *Tupiocoris notatus* feeding alters transcript levels of cytokinin inactivation and biosynthetic genes in attacked leaves of *Nicotiana attenuata*.

DOI: https://doi.org/10.7554/eLife.36268.010

**Figure supplement 2.** *Tupiocoris notatus* feeding on whole plants alters cytokinin (CKs) levels in attacked *Nicotiana attenuata* leaves.

DOI: https://doi.org/10.7554/eLife.36268.011

**Figure supplement 3.** Influence of *Tupiocoris notatus* single-leaf feeding on cytokinin levels of attacked and unattacked leaves.

DOI: https://doi.org/10.7554/eLife.36268.012

**Figure supplement 4.** *Tupiocoris notatus* contains large amounts of $N^6$-isopentenyladenine (IP) in their bodies independently of stage, sex or food source.

DOI: https://doi.org/10.7554/eLife.36268.013

contents gradually increased over time (*Figure 3b*). Levels of *cis*-zeatin (*cZ*) (*Figure 3—figure supplement 2a*), *cis*-zeatin riboside (*cZR*) (*Figure 3—figure supplement 2d*), *trans*-zeatin (*tZ*) (*Figure 3—figure supplement 2b*) and *trans*-zeatin riboside (*tZR*) (*Figure 3—figure supplement 2e*) were significantly higher after *T. notatus* attack. In contrast, levels of $N^6$-isopentenyladenine (IP) remained unaffected by mirid feeding (*Figure 3—figure supplement 2c*) and levels of $N^6$-isopentenyladenosine (IPR) decreased in attacked leaves (*Figure 3—figure supplement 2f*). This decrease was significant in the first 24 hr after the initiation of mirid attack and disappeared at later harvest times ($p < 0.05$ in TukeyHSD *post hoc* test).

When mirids were only allowed to feed on a single leaf, we did not observe changes in levels of summed CKs over the whole time series (*Figure 3—figure supplement 3b*). However, Bonferroni-corrected t-tests of single time point revealed increased levels at the last time point harvested, after 144 hr of feeding (tt: $p = 0.026$). The changes in individual CKs only partially overlapped with those observed during whole-plant feeding. There was a significant increase in *cZ* (*Figure 3—figure supplement 3c*), but levels of *cZR* decreased (*Figure 3—figure supplement 3f*). IP levels, which did not change during whole-plant feeding, were significantly higher overall after single-leaf feeding, although pairwise comparisons for each time point did not reveal significant changes at any given time point. *tZ*, *tZR* and IPR remained unaffected by mirid feeding when only single leaves were attacked (*Figure 3—figure supplement 3d,g,h*).

Interestingly, in both experiments overall CK levels remained unchanged or were increased, despite the concomitant increases in transcripts of genes related to CK degradation. From these results, we infer that CKs are involved in the observed nutritional stability during mirid feeding; this hypothesis prompted a more detailed analysis of the origin of these CKs.

### *T. notatus* contains high levels of IP

Mirid attack enhanced the levels of *cZ* and *cZR* as was previously found for *M. sexta* herbivory, wounding, and JA application (*Schäfer et al., 2015c*; *Brütting et al., 2017*); but in contrast to these other types of elicitations, long-term mirid feeding and the associated JA accumulation did not decrease IP levels. This was particularly surprising given that CK degradation and inactivation processes appeared to have been activated by mirid feeding. We analyzed CK levels in *T. notatus* to determine whether these insects could themselves provide a source of CKs. We found very high levels of IP and IPR in extracts from the insect bodies (*Figure 3c*). While concentrations of IPR were comparable to those in leaves (around 1 pmol per g fresh mass (FM)), concentrations of IP exceeded those of leaves by up to three orders of magnitude: while levels in leaves ranged from 0.01 to 0.1 pmol g $FM^{-1}$, levels in insects were usually between 1 and 5 pmol g $FM^{-1}$ and attained values as high as 16 pmol g $FM^{-1}$. Insects collected from *N. attenuata* plants in their natural habitat at a field site in Utah, USA, also contained high amounts of IP: in a pooled sample of ten insects, we measured 18.26 pmol IP per g $FM^{-1}$.

Mirids contained high IP and IPR levels in their bodies independently of their sex, developmental stage, or food source (*Figure 3—figure supplement 4*). The sole significant difference was that IP concentration in nymphs was about half as high as in adult males and females, but nymphs still had concentration several times those found in leaves (*Figure 3—figure supplement 4a*). To evaluate if CKs levels remained stable when *T. notatus* was no longer feeding on its host plant, we reared insects for five days either on artificial diet (containing no CKs) or on plants. Insects raised on artificial diet had IP levels in their body that were not different from levels in insects raised on plants; IPR levels were also unchanged (*Figure 3—figure supplement 4b*).

Although the source of CKs in *T. notatus* remains unknown, we hypothesize that IP and IPR found in *T. notatus* body could be used by the mirid to counter the decrease in IP levels in attacked leaves that is commonly observed in response to long-term JA elicitation or *M. sexta* feeding.

### *T. notatus* transfers IP to the plant via its oral secretions

To evaluate whether *T. notatus* could transfer CKs to the plant, we conducted $^{15}$N- labeling experiments. We grew plants in hydroponic culture with $^{15}$N-labeled $KNO_3$ as the only source of nitrogen. We furthermore created a stock of *T. notatus* insects that were $^{15}$N-labeled by raising them for an entire generation on $^{15}$N-grown plants. We then performed two different types of experiments to trace the origin of CKs in *T. notatus* attacked leaves: we either used $^{15}$N-grown plants that we

exposed to $^{14}$N-labeled insects (*Figure 4* and *Figure 4—figure supplement 1*) or we used $^{14}$N-grown plants and exposed them to $^{15}$N-labeled insects (*Figure 4—figure supplements 2* and *3*). CKs are adenine derivatives that contain five nitrogen atoms. Therefore, CKs produced by $^{15}$N-labeled plants or insects harbored five $^{15}$N and are readily distinguished from $^{14}$N-labeled CKs by mass spectrometry (*Figure 4—figure supplements 4* and *5*).

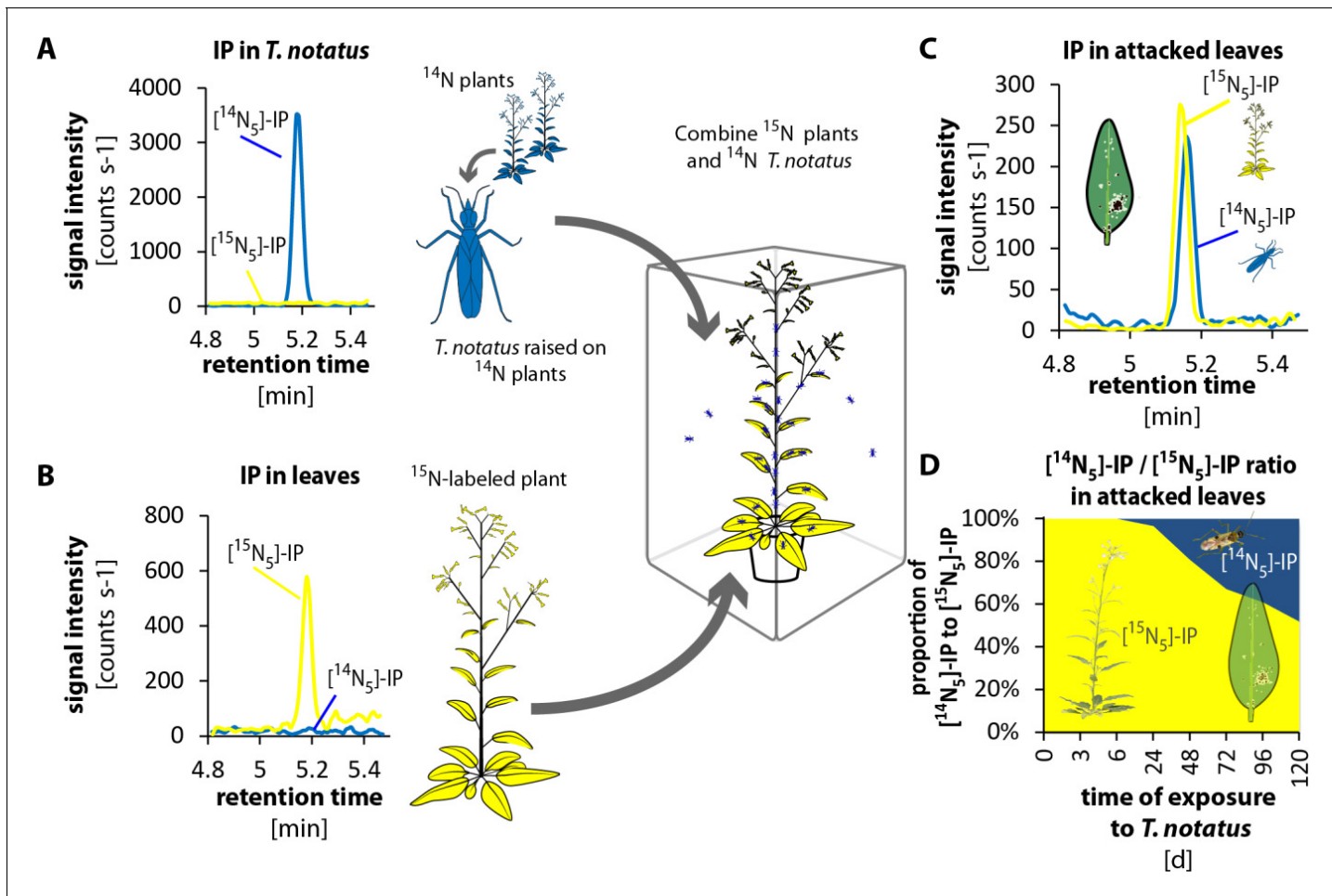

**Figure 4.** *Tupiocoris notatus* transfers IP to leaves of its host plant. (**A**) and (**B**) Experimental setup and chromatograms of IP: (**A**) *T. notatus* raised on $^{14}$N-grown hydroponic plants grown contain only [$^{14}$N$_5$]-IP in their bodies. (**B**) Plants raised on a hydroponic medium containing only a $^{15}$N containing N-source have only [$^{15}$N$_5$]-IP in leaves. $^{15}$N labeled plants and $^{14}$N labeled insects were placed in the same cage for 5 days. Ratio of [$^{14}$N$_5$]-IP (originating from insects, blue) and [$^{15}$N$_5$]-IP (from host plant, yellow) were determined in attacked leaves. (**C**) Chromatograms of [$^{14}$N$_5$]-IP and [$^{15}$N$_5$]-IP in the leaves of 5d attacked plants. (**D**) Ratio of [$^{14}$N$_5$]-IP and [$^{15}$N$_5$]-IP at different harvest times after the start of exposure to *T. notatus* (N = 5). For raw data see Raw_data_FIGURE_4 (Dryad: *Brütting et al., 2018*).

DOI: https://doi.org/10.7554/eLife.36268.014

The following figure supplements are available for figure 4:

**Figure supplement 1.** *Tupiocoris notatus* transfers IPR to leaves of its host plant.
DOI: https://doi.org/10.7554/eLife.36268.015
**Figure supplement 2.** Twenty *Tupiocoris notatus* individuals transfer detectable amounts of IP to leaves.
DOI: https://doi.org/10.7554/eLife.36268.016
**Figure supplement 3.** Twenty *Tupiocoris notatus* individuals transfer detectable amounts of IPR to leaves.
DOI: https://doi.org/10.7554/eLife.36268.017
**Figure supplement 4.** Chromatograms of IP, [D$_6$]-IP, [$^{15}$N$_5$]-IP.
DOI: https://doi.org/10.7554/eLife.36268.018
**Figure supplement 5.** Chromatograms of IPR, [D$_6$]-IPR and [$^{15}$N$_5$]-IPR.
DOI: https://doi.org/10.7554/eLife.36268.019

In the first approach, we used a low-infestation setup by placing 20 $^{15}$N-labeled *T. notatus* adults in a small cage on the leaf of a $^{14}$N-grown plant for five days. After four days of continuous feeding, we found detectable amounts of $^{15}$N-labeled IP (and IPR) in the leaves (*Figure 4—figure supplements 2* and *3*): around 2.35 fmol [$^{15}$N$_5$]-IP per g FM, which represent the 3.3% of the [$^{15}$N$_5$]/[$^{14}$N$_5$]-IP ratio (*Figure 4—figure supplement 2d*). [$^{15}$N$_5$]-labelled IP and IPR could only have originated from the insects, as the natural abundance of $^{15}$N is below 0.4%, and IP (or IPR) with five $^{15}$N would occur about once in a trillion molecules. From these values, one mirid feeding on a leaf for five days could account for a transfer of at least 0.12 fmol IP per g FM$^{-1}$ (*Supplementary file 1*), assuming that CK transport, degradation or conversion to other CK forms can be excluded.

In the reverse experiment, we used $^{15}$N-grown plants and insects raised on $^{14}$N-grown plants (*Figure 4* and *Figure 4—figure supplement 1*). We placed $^{15}$N-grown plants in cages where *T. notatus* were reared on $^{14}$N-grown plants. These $^{15}$N-grown plants were switched to a new cage with infested $^{14}$N-grown plants once per day to ensure that they were always attacked by $^{14}$N-labeled insects and to prevent the accumulation of $^{15}$N in the $^{14}$N-labeled insects. After 5 days, an average of 48% of the [$^{15}$N$_5$]/[$^{14}$N$_5$]-IP ratio was $^{14}$N labeled and therefore originating from the insects (*Figure 4*). In this stronger induction setup, IPR transfer from the insect to the plant was also already detected after 24 hr and accounted to 19% of the [$^{15}$N$_5$]/[$^{14}$N$_5$]-IP ratio after 5 d (*Figure 4—figure supplement 1*).

To evaluate how IP and IPR were transferred to the leaf during feeding, we analyzed the CK contents of the oral secretions and frass of *T. notatus*, which we considered the most likely means of transfer. Mirids were fed on sugar solutions covered with parafilm, which allowed the insects to penetrate the film with their stylets while preventing evaporation and preventing either insects or their frass from being immersed in the liquid. We then measured CKs in the sugar solution, which contained substances transferred by the oral secretions, as well as in the surface wash, which contained insect excretions (frass). We found large amounts (high signal intensity) of IP mainly from the oral secretions (from the sugar solution that mirids had fed on) and much lower amounts in the frass of the mirids (from the surface wash) (*Figure 5*). IPR was found in oral secretions and in frass in similar amounts (*Figure 5—figure supplement 1*).

These results clearly demonstrate that *T. notatus* is able to transfer CKs (mainly IP) to its host plant. The most likely means of transfer would be via the salivary secretion produced during feeding, although we cannot rule out a smaller contribution of feces, which are sticky and tend to cover infested leaves.

## Altered CK metabolism in *N. attenuata* affects its interaction with *T. notatus*

In nature, *T. notatus* feeds on young *N. attenuata* tissues, such as younger stem leaves and young growing leaves. This feeding pattern was inferred from the damage distributions observed on plants in both nature and the glasshouse (*Figure 6—figure supplement 1a*), as well as in two-choice assays (*Figure 6—figure supplement 1b*). The young leaves preferred by *T. notatus* are typically rich in CKs (*Brütting et al., 2017*). To evaluate how CK metabolism affects the interaction of *N. attenuata* with *T. notatus*, we used transgenic *N. attenuata* plants that were either enhanced in CK production (i-ov*ipt*) or silenced in CK perception (ir*chk2/3*). Transgenic i-ov*ipt* plants that contain a dexamethasone (DEX)-inducible promotor system coupled to an IPT gene were produced as previously described (*Schäfer et al., 2013*; *Schäfer et al., 2015b*), and allowed a DEX-mediated induction of CK overproduction. ir*chk2/3* plants, fully characterized in *Schäfer et al., (2015b)* are silenced for two of three CK receptors.

*T. notatus* prefers leaves of i-ov*ipt* plants which have been treated with DEX and therefore have higher levels of CKs (*Figure 6a*). If *T. notatus* is given the choice between empty vector (EV) and ir*chk2/3* plants, mirids show a strong preference for EV plants, as shown in the lower damage levels on ir*chk2/3* plants (*Figure 6b*). Furthermore, we found pronounced differences in the reaction of the plants to the damage caused by *T. notatus* feeding. Mirid attack caused necrotic lesions in ir*chk2/3* plants, comparable to a pathogen-induced hypersensitive response, whereas this did not occur in WT, EV or i-ov*ipt* plants (*Figure 6c*).

To better understand the feeding preferences of *T. notatus*, we measured nutrient levels in ir*chk2/3*, DEX-induced i-ov*ipt* plants and EV plants (*Figure 7*). Starch and sucrose did not differ among the lines (*Figure 7c,f*). However, i-ov*ipt* plants had higher concentrations of protein, free

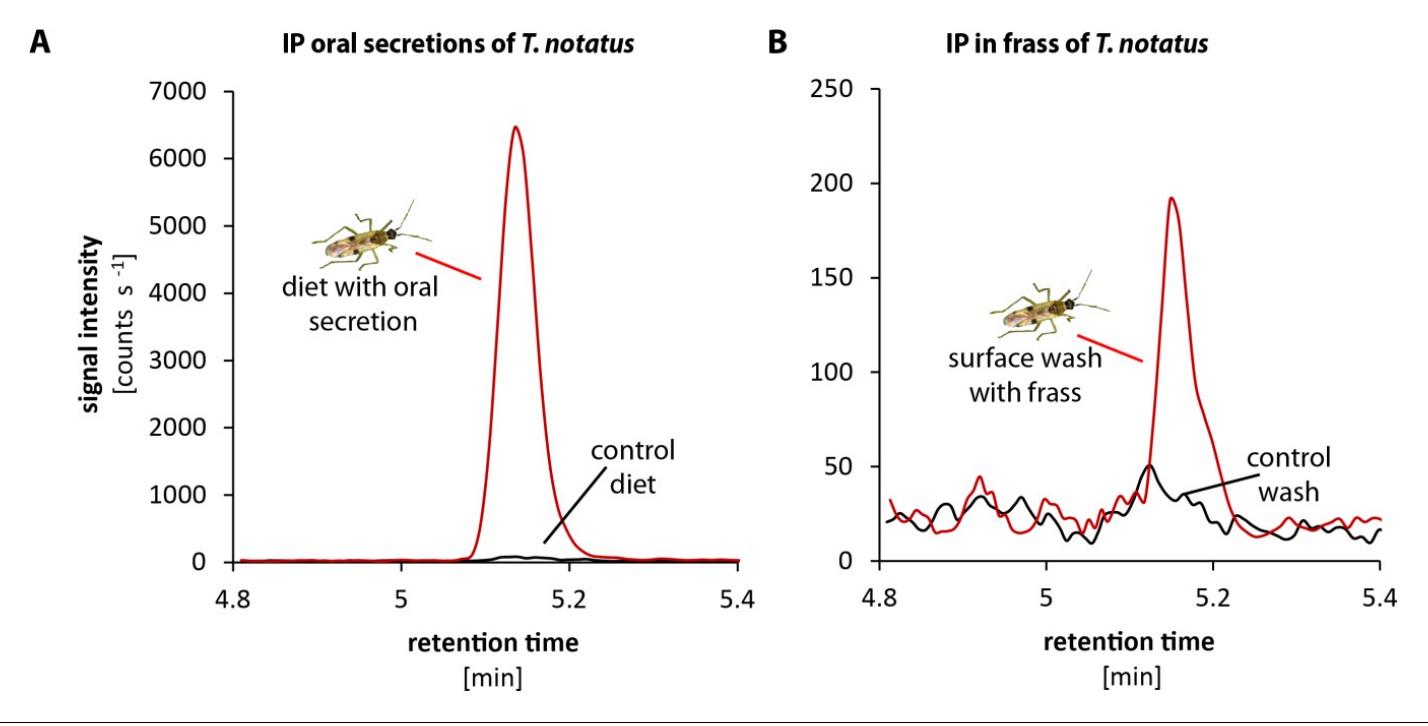

**Figure 5.** *Tupiocoris notatus* contain large quantities of IP in their saliva and small amounts in their frass. Chromatograms showing the signal intensity of an MS/MS- trace for IP (204.1 → 136.0). (**A**) IP signal of pure sugar solution (black line) or sugar solution fed upon by *T. notatus* for 5 days (red line). The sugar solution was covered with a thin layer of parafilm that allowed piercing and feeding on the solution and prevented contamination by *T. notatus* frass. (**B**) Chromatograms of the surface wash of the parafilm covering the sugar solution after *T. notatus* feeding (red line, covered with visible frass spots) or without (control wash, black line). Chromatograms shown represent one out of six replicates.

DOI: https://doi.org/10.7554/eLife.36268.020

The following figure supplement is available for figure 5:

**Figure supplement 1.** *Tupiocoris notatus* contain IPR in their saliva and frass.

DOI: https://doi.org/10.7554/eLife.36268.021

amino acids, glucose and fructose than did ir*chk2/3* plants (**Figure 7a,b,d,e**). The i-ov*ipt* plants tended to have higher nutrient levels than did EV plants but the results were only statistically significant for glucose concentrations (**Figure 7d**). In contrast, ir*chk2/3* plants tended to have lower nutrient levels compared to EV, but these were only significantly lower for fructose concentrations (**Figure 7e**).

From these results, we conclude that CKs play a dual role in the *T. notatus-N. attenuata* interaction: as important determinants of tissue palatability for *T. notatus* by enhancing nutrient contents, but also as important tolerance factors that allow plants to suffer negligible or lower fitness consequences of mirid attack than they would otherwise.

## Discussion

Gall-formers and leaf-miners have long been known for their ability to manipulate plant's physiology, likely via phytohormones such as CKs. It is commonly thought that CK-dependent manipulation of plant metabolism is a trait typical of endophytic insects that has been shaped by the sedentary and intimate relationships that these insects establish with their host plants (**Giron et al., 2016**). Here, we provide evidence that a free-living insect, the mirid *T. notatus,* transfers CKs to its host plant *N. attenuata* and likely manipulates the host plant's metabolism for its own benefit. This strategy, not previously described for any free-living insects, indicates that CK-mediated manipulation of plant metabolism by insects could be a mechanism more widespread than previously thought.

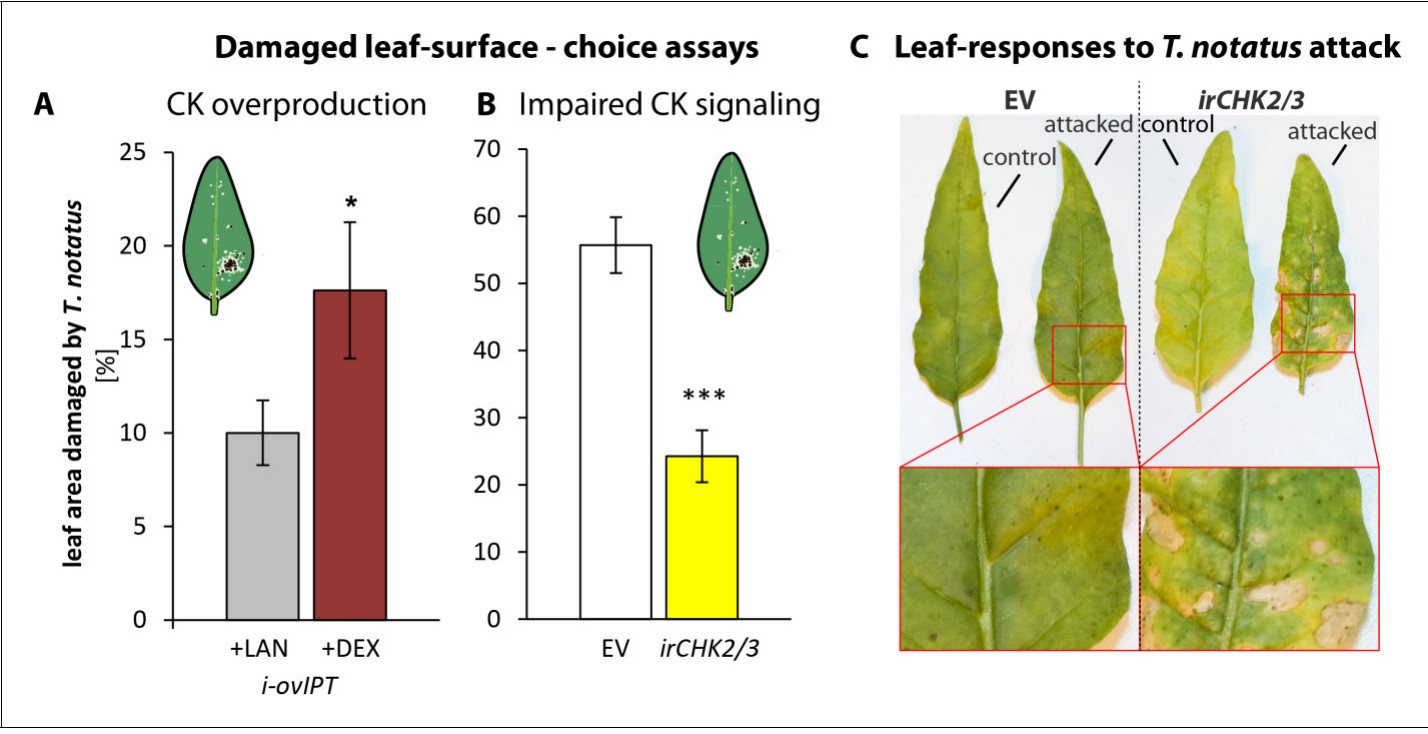

**Figure 6.** Cytokinin-regulated traits mediate *Tupiocoris notatus* feeding preferences and alter leaf responses to feeding. (**A**) and (**B**): Surface damage on *N. attenuata* plants after 10 d of *T. notatus* feeding. (**A**) *T. notatus* could choose between dexamethasone-inducible isopentenyltransferase-overexpressing plants (i-ov*ipt*) treated with dexamethasone-containing lanolin paste (+DEX) or lanolin paste without dexamethasone as control (+LAN; figure based on data from *Schäfer et al., 2013*). Statistically significant differences were identified with pairwise t-test: N = 7, p=0.032. (**B**) Choice between empty vector (EV) and ir*chk2/3* plants silenced in the two cytokinin receptor genes *NaCHK2* and *NaCHK3* (ir*chk2/3*). Pairwise t-test: N = 6, p<0.001. Error bars depict standard errors. *p<0.05, ***p<0.001. (**C**) Representative pictures of leaves of EV or ir*chk2/3* plants with or without *T. notatus* damage. Magnifications show necrotic lesions occurring only in ir*chk2/3* plants after several days of mirid feeding. For raw data see Raw_data_FIGURE_6 (Dryad: *Brütting et al., 2018*).

DOI: https://doi.org/10.7554/eLife.36268.022

The following figure supplement is available for figure 6:

**Figure supplement 1.** *Tupiocoris notatus* prefers to feed on young leaves.

DOI: https://doi.org/10.7554/eLife.36268.023

## Transfer of CKs from *T. notatus* to its host plant

We unambiguously demonstrated that *T. notatus* transfers two types of CKs, IP and its riboside IPR, to *N. attenuata* providing the first clear demonstration of CK transfer from an insect to a plant. IP has been generally considered one of the most active natural CKs based on classical activity assays (*Gyulai and Heszky, 1994*; *Sakakibara, 2006*) and high concentrations of IP have been previously reported in plant-manipulating endophytic insects, like leaf miners and gall-formers (*Engelbrecht, 1968*; *Engelbrecht et al., 1969*; *Mapes and Davies, 2001*; *Straka et al., 2010*; *Yamaguchi et al., 2012*; *Body et al., 2013*; *Tanaka et al., 2013*). We also showed that oral secretions, and in much lower amounts, frass of *T. notatus*, contained IP and IPR, thus providing a possible means of transfer.

Concentrations of total CK content of *N. attenuata* leaves steadily increased during long-term *T. notatus* feeding, consistent with the observation that mirids transferred CKs to plants. The overall reconfiguration of the transcriptional activity of genes involved in CK degradation, inactivation, and biosynthesis upon *T. notatus* feeding did not correlate with the apparent changes in CK concentrations. This suggests that *N. attenuata* might activate a type of CK detoxification in response to mirid feeding and CK introduction. Additional support for the existence of a mechanism that counter-balances mirid-injected CKs comes from the observation that the leaf concentration of IP and IPR, the two CKs transferred by *T. notatus*, were unaffected or only slightly changed in plants by mirid

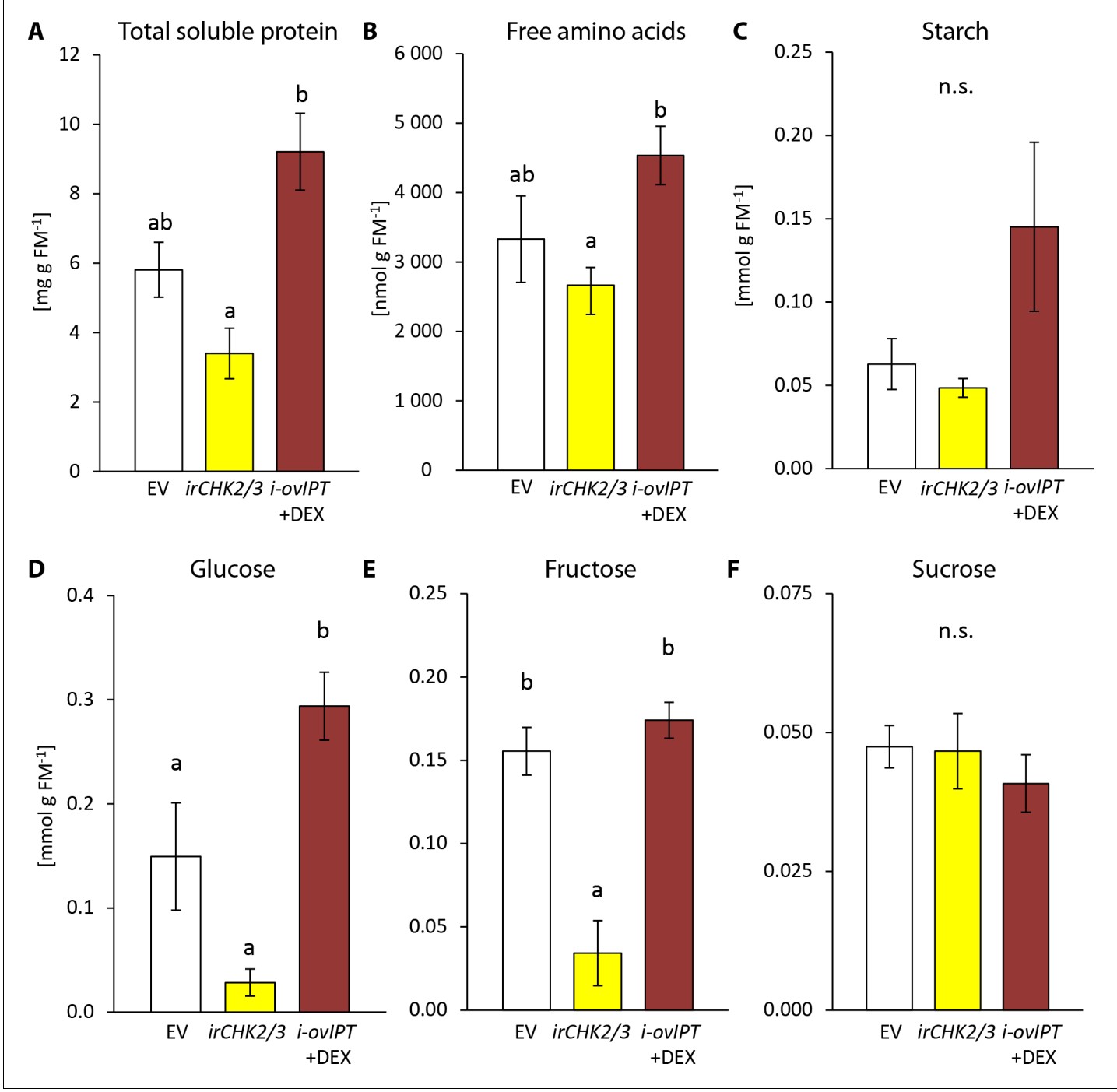

**Figure 7.** Transgenic *Nicotiana attenuata* plants altered in their cytokinin metabolism are also altered in their nutrient contents. We compared nutrient contents of empty vector (EV) plants, plants silenced in the two cytokinin receptor genes *NaCHK2* and *NaCHK3* (ir*chk2/3*) and dexamethasone-inducible isopentenyltransferase-overexpressing plants (i-ov*ipt*) treated with dexamethasone-containing lanolin paste (DEX) leading to spatially-regulated increased CK levels. Concentrations were determined in untreated rosette leaves of *N. attenuata:* (**A**) protein, (**B**) free amino acids, (**C**) starch, (**D**) glucose, (**E**) fructose and (**F**) sucrose. Significant differences were identified with one-way ANOVAs followed by Tukey HSD *post hoc* tests. (**A**) protein: $F_{2,9}$ = 10.74, p=0.004; (**B**) free amino acids: $F_{2,9}$ = 4.27, p=0.050; (**C**) $\log_2$ transformed starch: $F_{2,9}$ = 2.208, p=0.166; **D**) glucose: $F_{2,9}$ = 18.89, p<0.001; (**E**) fructose: $F_{2,9}$ = 15.43, p=0.001; (**F**) sucrose: $F_{2,9}$ = 0.375, p=0.698. Error bars depict standard errors (N = 4). FM: fresh mass. For raw data see Raw_data_FIGURE_7 (Dryad: ***Brütting et al., 2018***).

DOI: https://doi.org/10.7554/eLife.36268.024

feeding. This was surprising, considering that we estimated that after five days of whole-plant infestation, roughly half of the total IP in attacked leaves originated from mirids.

Understanding to what extent the observed changes in cytokinin signaling result from mirid-mediated CK transfer is further complicated by the fact that CK levels respond as part of the herbivory-inducible defense signaling (*Schäfer et al., 2015a*; *Schäfer et al., 2015c*; *Brütting et al., 2017*). The dual role of CKs in plant growth and defense highlights the complexity of the fine regulation of CKs needed to regulate plant physiological responses. Not only CK quantities, but also CK structures, and the hormonal balance with other phytohormones may influence changes in metabolism upon insect feeding (*Giron et al., 2013*).

## *T. notatus*-induced effects in *N. attenuata*

Demonstrable effects on host plants induced by endophytic insects include alterations of plant morphology, changes in the nutritional quality of the affected tissues and the inactivation of plant defenses surrounding the attack sites (*Giron et al., 2016*). Whereas alterations of plant morphology are associated with only some endophytic insects, for example gall-formers, control of the nutritional quality of the infested tissues seems to be a common feature of all endophytic insect-plant interactions.

*N. attenuata* leaves maintain their nutritional quality despite being heavily damaged by *T. notatus* feeding: only total soluble proteins (TSPs) decreased with heavy infestation, as in the whole-plant experiments, whereas concentrations of glucose, fructose, sucrose and starch remained unchanged. Previous studies in *N. attenuata* showed that wounding and application of oral secretions (OS) of *M. sexta* as well as *M. sexta* feeding reduced glucose and fructose concentrations by inhibiting soluble invertases. Such reductions are JA-dependent, and abrogated in transgenic lines impaired in JA production (*Machado et al., 2015*). A negative influence of jasmonates on plant primary metabolism has also been suggested by studies in a number of other plants (*Babst et al., 2005*; *Skrzypek et al., 2005*; *van Dam and Oomen, 2008*; *Hanik et al., 2010*; *Tytgat et al., 2013*). Hence, the fact that *T. notatus* feeding activated JA-signaling, but did not negatively influence soluble monosaccharide concentrations, suggested that an additional counterbalancing alteration in the primary metabolism of *N. attenuata* occurs during *T. notatus* feeding. Similar to what has been observed for carbohydrates, wounding and *M. sexta* OS application results in a 91% reduction of total soluble proteins (TSPs) in young rosette leaves (*Ullmann-Zeunert et al., 2013*), the same leaf stage used in this study. After 144 hr of continuous *T. notatus* infestation, during which proteins should be heavily depleted by mechanical cell-content damage – in contrast to the minor damage associated with OS elicitation (*Halitschke et al., 2001*) – TSP reductions were only ca. 75%. More surprisingly, a smaller *T. notatus* infestation (twenty mirids confined on a single leaf) did not change TSP contents at all during 144 hr of continuous feeding. These results are consistent with microarray analysis that compared expression patterns induced by *T. notatus* and *M. sexta*, which revealed that mirid-specific transcriptional responses occurred largely in primary metabolism (*Voelckel and Baldwin, 2004*). Thus, during *T. notatus* feeding, plant's primary metabolism seems to be influenced by mechanisms different from the classical JA-mediated herbivory and wound responses.

In contrast to the apparent sugar and starch homeostasis observed during *T. notatus* feeding and to the observations of *Halitschke et al. (2011)*, we observed a decrease in photosynthetic rates during continuos mirid feeding. Reduced photosynthetic rate is a general response observed in a number of plant-insect interactions (*Zhou et al., 2015*) as well as in *N. attenuata*. Wounding and elicitation with *M. sexta* OS rapidly decrease photosynthetic $CO_2$ assimilation and this reduction is mediated by the JA-precursor OPDA (*Meza-Canales et al., 2017*). We think that the discrepancy between our work and results from *Halitschke et al. (2011)* likely results from differences in the experimental protocols used: (1) we used a very heavy infestation, (2) the leaf area used to measure the photosynthetic rates of mirid-attacked leaves included both damaged and undamaged areas, (3) we did not normalize photosynthetic rates to intact undamaged leaf tissue. In any case, the reduction in the overall photosynthetic rates observed during *T. notatus* feeding was not consistent with unchanged starch and sugar levels and, together with the observation that *T. notatus* transfers CK during feeding, suggests the inhibition of senescence and/or transport of nutrients to the attacked leaves.

Manipulation of plant defenses is another phenomenon often observed in endophytic insect-host plant interactions (*Giron et al., 2016*). We showed that *T. notatus* feeding activated *N. attenuata*

JA-dependent defense pathways in a way consistent with previous studies; the increases in defense metabolites induced by *T. notatus* feeding were comparable to those elicited by *M. sexta* attack (*Kessler and Baldwin, 2004*). *T. notatus* is well-adapted to the specialized metabolism of *N. attenuata*; it prefers wild-type plants which are less susceptible to invasion by other herbivores, rather than those impaired in JA biosynthesis with reduced defense metabolites (*Fragoso et al., 2014*). Insects counter the presence of toxic metabolites in their host plants by detoxification or sequestration of toxic substances (*Heckel, 2014*), and the detoxification ability of *T. notatus* is suggested by the observation that it accumulates transcripts encoding detoxification enzymes in response to JA-dependent defenses (*Crava et al., 2016*). This fact, togheter with the finding that defense pathways were not down-regulated during the *T. notatus* feeding, point out that down-regulation of plant defense may not benefit *T. notatus* as much as its manipulation of its host's nutritional status.

## Changes in the CK metabolism of the host plant alters *T. notatus* feeding preferences

Choice assays demonstrated that *T. notatus* is attracted to plant tissues with enhanced CK levels, both when CK levels were naturally higher, as in young plant tissues, and when CKs are experimentally increased using DEX-inducible transgenic plants (*Schäfer et al., 2013*). When CK perception was impaired as in the ir*chk2/3* line, mirids preferred WT or EV plants over the transgenic plants as shown by their different damage levels. This preference for higher CK levels and against ir*chk2/3* plants could either be a direct effect of CKs or – more likely – an indirect effect of CK-related processes. A direct attraction to CKs in insects has been discussed (*Robischon, 2015*) but to our knowledge there is no direct evidence that insects perceive CKs. Consistent with the second hypothesis, CK levels were not reduced in ir*chk2/3* plants compared to those of the EV line (*Schäfer et al., 2015c*). Thus, we infer that *T. notatus* prefers metabolites positively associated with the CK pathway. These might be molecules produced either by *N. attenuata* primary or specialized metabolism. For example, *T. notatus* is attracted to quercetin (*Roda et al., 2003*), and some related phenolic compounds are influenced by CK levels (*Schäfer et al., 2015b*; *Brütting et al., 2017*). Consistent with the primary metabolism hypothesis, we showed that nutrient levels of transgenic lines correlated with *T. notatus* preference. This inferred preference for nutrients is also consistent with *T. notatus* damage distribution on whole plants, which is concentrated on young, CK-rich and nutrient-rich tissues. These tissues are also better defended (*Brütting et al., 2017*) suggesting a possible trade-off between palatability and anti-digestive effects of the diet. However, specialized detoxification mechanisms likely allow *T. notatus* to feed with impunity on otherwise well-defended tissues (*Crava et al., 2016*), thus allowing *T. notatus's* feeding choice to reflect the nutritional quality, rather than the defensive status, of its host.

## Manipulation of the host plant physiology: a mechanism shared with free-living insects?

Our results provide evidence that a free-living insect transfers CKs which manipulates its host plant's metabolism, likely for its own benefit. CK-mediated plant manipulation strategies have only been known so far from endophytic insects (*Giron et al., 2016*). The low mobility and intimate associations of endophytic insects with their host plants provides the selective environment for the evolution of mechanisms that allow them to manipulate host plant physiology and/or morphology.

Species known for CK-dependent manipulation of host plants are not closely related to each other, and span several orders: Lepidoptera, Hymenoptera, Hemiptera and Diptera. The most studied examples are Lepidopteran leaf-miners which cause the green island phenomenon, like *Phyllonorycter blancardella* (*Giron et al., 2007*; *Kaiser et al., 2010*; *Body et al., 2013*; *Zhang et al., 2017*) or *Stigmella argentipedella* (*Engelbrecht, 1968*; *Engelbrecht, 1971*; *Engelbrecht et al., 1969*). Among gall-forming organisms, two species of the genus *Bruggmannia* are also capable of producing green islands (*Fernandes et al., 2008*). Other gallers that manipulate plant morphology can be found among hymenopterans, such as gall-wasps, dipterans such as gall-midges and gall-flies, and hemipterans such as psyllids and gall-aphids. Among these, a role for CKs in gall formation has been shown for the dipterans *Eurosta solidaginis* (*Mapes and Davies, 2001*) and *Rhopalomyia yomogicola* (*Tanaka et al., 2013*), the hymenopteran *Dryocosmus kuriphilus* (*Matsui et al., 1975*) and sawflies of the genus *Pontania* (*Yamaguchi et al., 2012*) and the hemipterans *Pachypsylla*

*celtidis* (*Straka et al., 2010*) and the galling-aphid *Tetraneura nigriabdominalis* (*Takei et al., 2015*). The fact that species from different orders have developed similar mechanism of plant manipulation suggests either an ancient evolutionary origin or a convergent evolutionary trait. We propose that such CK-dependent manipulation is more widespread than previously thought, and is also shared with free-living insects like *T. notatus*.

CKs transferred by *T. notatus* could originate from its host plant and be sequestered by the insect but also they could be synthesized by the insect itself or its associate endosymbionts. In fact, CKs are also produced by organisms other than plants, like fungi (*Chanclud et al., 2016*), bacteria (*Costacurta and Vanderleyden, 1995*) and nematodes (*Siddique et al., 2015*). It is thought that IP and IPR can be derived from tRNA, and this suggests that the substrate for CK biosynthesis is shared by all organisms (*Persson et al., 1994*), virtually enabling insects to produce CKs. The most recent studies on CK-mediated manipulation of plant physiology by insects suggested a role of endosymbiotic bacteria in CK production (*Kaiser et al., 2010*; *Giron et al., 2013*; *Zhang et al., 2017*). Antibiotic feeding experiments have revealed that endosymbionts like *Wolbachia* are the most likely producers of CKs in the leaf-miner *P. blancardella* (*Kaiser et al., 2010*; *Body et al., 2013*). Yet, this bacterium is unlikely responsible for CK production in *T. notatus*, as *Wolbachia* was not be detected in mirids from the same glasshouse colony used in our experiments, and was identified only very rarely from insects collected from the field (*Adam et al., 2017*).

## Conclusions

Free-living phytophagous insects are thought not to manipulate their host plant's physiology to enhance the nutritional quality of their diet, as they are free to move to the best feeding locations on a plant. This work provides evidence of the ability of a free-living insect to introduce CKs into their host during feeding to maintain a better nutritional environment. We suggest that this mechanism may be commonly found in other free-living species and that it combines the benefits of the two different lifestyles: the ability to move, hide and choose the best feeding locations, and to manipulate the host plant via CK-transfer. Clarifying the details of the origins of the *T. notatus*-transferred CKs and studying their role in nature will provide new insights into the complex interactions that occur during plant-herbivore interactions.

## Materials and methods

### Chemicals

All chemicals used were obtained from Sigma-Aldrich (St. Louis, MO, US) (http://www.sigmaaldrich.com/), Merck (Darmstadt, Germany) (http://www.merck.com/), Roth (Karsruhe, Germany) (http://www.carlroth.com/) or VWR (Radnor, PE, US) (http://www.vwr.com), if not mentioned otherwise in the text. CK standards were obtained from Olchemim (Olomuc, Czech Republic) (http://www.olchemim.cz), dexamethasone (DEX) from Enzo Life Sciences (Farmingdale, NY, US) (http://www.enzolifesciences.com/), HCOOH for ultra-performance LC from Fisher Scientific (Hampton, NH, US) (http://www.fisher.co.uk/) or from Honeywell Riedel-de Haën™ (Morris Plains, NJ, US) (http://www.riedeldehaen.com/), and GB5 from Duchefa (Haarlem, The Netherlands) (http://www.duchefa-biochemie.nl/).

### Plant cultivation and transgenic plants

We used the 31st inbred generation of *Nicotiana attenuata* (Torr. ex S. Wats.) originating from the 'Desert Inn' population from the Great Basin Desert (Washington County, UT, US) as wild-type (WT) plants (*Baldwin et al., 1994*). Transgenic plants were generated from WT *N. attenuata*, as described by *Krügel et al., 2002*) by *Agrobacterium*-mediated transformation. Empty vector transformed plants (EV) (line A-04-266-3) were used as controls in experiments that included other transgenic lines.

The transgenic line ir*chk2/3* was transformed with a construct harboring inverted-repeat gene fragments to silence the expression of two of the three known CK receptor homologs *NaCHK2* and *NaCHK3* (Chase Domain Containing Histidine Kinase 2 and 3) and was previously characterized (*Schäfer et al., 2015b*). We used line A-12–356, which had a silencing efficiency of about 50%. The i-ov*ipt* line (A-11–92 x A-11–61) contains a gene encoding the rate-limiting step of CK biosynthesis,

the isopentenyltransferase (IPT) from *Agrobacterium tumefaciens* (*Tumor morphology root; Tmr*). The IPT gene is controlled by the pOp6/LhGR expression system, which allows transcriptional up-regulation in a specific tissue by the application of DEX (*Schäfer et al., 2013*). Application of DEX to the leaves of the plant induces the transcription of *IPT* which locally increases CK levels. DEX was dissolved in lanolin paste with 1% DMSO at a final concentration of 5 μM. For control treatments, we used 1% DMSO in lanolin. The lanolin paste was applied to the petioles of the leaves 24 hr prior to other treatments as previously described (*Schäfer et al., 2013*).

Seeds were sterilized and germinated on Gamborg B5 media as described by *Krügel et al. (2002)* with modifications as previously described (*Brütting et al., 2017*). For soil growth conditions, ten days after germination, plants were first transplanted to TEKU pots and 10 days later into 1 L pots. For hydroponic growth conditions, the plants were transferred after 12 days into 50 mL hydroponic culture single pots and 10 days later into 1 L hydroponic containers. Conditions for hydroponic culture were previously described (*Ullmann-Zeunert et al., 2012*) as were conditions for soil growth (*Brütting et al., 2017*). For the damage determination experiment in the glasshouse, single plants were grown in 4 L pots. Plants were maintained under glasshouse conditions (22–27°C; ca. 60% RH, 16:8 light:dark regime) as previously described (*Adam et al., 2017*).

To prevent *T. notatus* infestation of the main glasshouse facility of the MPI-COE, we maintain the colony in a separate glasshouse located in Isserstedt, Germany, approximately 7 km from the main glasshouse facility. Plants were germinated in the main glasshouse facility and transferred to the Issersted glasshouse just before plants began to flower, when plants had main stems about 25 cm tall. In both glasshouses, plants were maintained at comparable growth conditions. After transferring plants to the Isserstedt glasshouse, we allowed for at least two days of acclimation before initiating experiments.

## Insect colony

The colony of *T. notatus* (Distant, 1893) (FIGURE 1,1A) originated from insects caught in the vicinity of the Brigham Young University/Max Planck field station at Lytle Ranch Preserve in the Great Basin Desert (Washington County, UT, US) and was annually refreshed with insects caught from the same field site. The colony was maintained in cages made of acrylic glass (2 × 1 × 1 m) with a fine mesh for air circulation. Cages were maintained under the same glasshouse growth conditions that were used for the cultivation of *N. attenuata* (27°C; ca. 60% RH, 18:8 light:dark regime). We fed insects with hydroponically grown WT *N. attenuata* plants. Fresh plants were provided weekly and remained in the cages for several weeks to allow nymphs to hatch from eggs laid in the older plants. Insects were collected from the cage for experiments using an insect exhauster. Prior to being clip-caged onto *N. attenuata* leaves, insects were anaesthetized with $CO_2$.

## *T. notatus* rearing on artificial diet

For the artificial diet we dissolved amino acids (L-alanine, 50 mg; L-arginine, 30 mg; L-cysteine, 20 mg; glycine, 20 mg; L-histidine, 30 mg; L-leucine, 30 mg; L-lysine, 20 mg; L-phenylalanine, 30 mg; L-proline, 80 mg; L-serine, 100 mg; L-tryptophan, 500 mg; L-tyrosine, 10 mg; L-valine, 40 mg; L-asparagine, 200 mg; L-aspartic acid, 200 mg; L-glutamine, 500 mg; L-glutamic acid, 300 mg; L-isoleucine, 20 mg; L-methionine, 10 mg; L-threonine, 100 mg), sugars (glucose, 400 mg; fructose, 150 mg; sucrose, 800 mg) and vitamins (Vanderzant Vitamin Mix, 650 mg) in 40 mL water and sterile filtered the solution. Additionally, we prepared an agar solution (1 g Agar-Agar in 60 mL water) which was sterilized by autoclaving. After cooling the liquid agar solution to approximately 60°C in a water bath, we added the nutrient solution and aliquoted the diet under sterile conditions in single 0.5 mL microcentrifuge tubes, where it solidified. These tubes were stored at 4°C until use.

For experiments, *T. notatus* were placed in plastic boxes (10 × 6 × 6 cm) covered with paper tissue and sealed with a perforated lid. In each box, 15 to 20 mirids were placed with a tube containing the artificial diet as the sole food source in addition to a source of water. The diet was exchanged with fresh diet every day. The shaded boxes were kept in the glasshouse. After 5 days the surviving mirids were collected, flash-frozen in liquid nitrogen and stored at −80°C until CK extraction.

## Collection of oral secretions and frass of *T. notatus*

*T. notatus* oral secretions were collected as previously described (*Halitschke et al., 2011*) with minor modifications. In brief, we placed 15 to 20 mirids in a single plastic box (10 × 6 × 6 cm) covered with paper tissue and sealed with a perforated lid. In each box, we placed an inverted scintillation vial lid filled to the brim with sugar solution (~3 mL, 40 mM glucose) as the sole food and water source. Lids were covered by a stretched thin layer of Parafilm (Neenah, WI, US) which allowed for stylet penetration. After 24 hr, we collected the lids, removed the sugar solution with a syringe and carefully dissolved (with MeOH) the frass spots deposited on the parafilm. As a control, similarly packaged sugar solutions in boxes lacking mirids maintained under the same conditions were used. Frass and sugar solution samples originating from the exposure to approximately 100 mirids were pooled. Pooled sugar solutions were freeze-dried overnight. Prior to CK extraction, extraction buffer was used to dissolve the evaporated sugar solution.

## Herbivory treatment

To measure the defense responses of *N. attenuata* to *T. notatus* feeding, whole plants were exposed to mirid attack in the *T. notatus* rearing cages and damaged lamina from the first (lowest) stem leaf were collected after three days of feeding. Control plants were placed in a similar but mirid-free cage under same conditions. Collected lamina pieces were flash-frozen in liquid nitrogen and stored at −80°C until analysis.

For kinetic analysis of CKs, JA, JA-Ile, primary metabolites and photosynthetic rates during *T. notatus* attack, we used two different experimental setups which differed in the area damaged by *T. notatus*, and the number of *T. notatus* used to inflict the damage to the *N. attenuata* plants. In the first setup, only one leaf per plant was exposed to *T. notatus.* We enclosed twenty adults on the first (lowest) stem leaf in a round plastic clip-cage (7 cm dimeter, 5 cm height). Clip-cages had holes covered with a fine mesh for air ventilation. Control plants received empty clip-cages to control for the effects of caging leaves. We also sampled leaves from plants without clip-cages. Before sampling, mirid mortality was scored, and samples with more than 50% mortality were discarded. Control and damaged leaf lamina were harvested at nine time-points from separate plants (0, 1, 3, 24, 48, 72, 96, 120 and 144 hr), flash-frozen in liquid nitrogen and kept at −80°C until analysis. In the second setup, the entire aboveground plant was exposed to mirids, by placing the plants directly into the *T. notatus* rearing cage; control plants were placed in a similar, empty cage. Damaged lamina from the first (lowest) stem leaf were sampled at seven time-points from separate plants (0, 24, 48, 72, 96, 120 and 144 hr). Both experiments were started in the morning (09:00 – 12:00) and each harvest time represented at least three replicate plants. For each experiment, phytohormone concentrations (CKs, JA and JA-Ile), sugars (sucrose, fructose and glucose), starch, total soluble proteins, photosynthetic rates and chlorophyll contents were measured.

## Measurement of caffeoylputrescine and nicotine

Caffeoylputrescine and nicotine were extracted and determined by UHPLC-ToF-MS by analyzing extracted ion chromatograms as previously described (*Schäfer et al., 2015c*; *Brütting et al., 2017*). For extraction, 80% MeOH (v/v) was used for approximately 100 mg of frozen and ground leaf material from each sample (exact tissue masses were recorded). Values are presented as peak area * g $FM^{-1}$.

## Trypsin proteinase inhibitor (TPI) assay

TPI activity was determined using a radial diffusion assay (*Jongsma et al., 1994*; *van Dam et al., 2001*) with approximately 50 mg of frozen and ground leaf lamina (exact tissue masses were recorded). TPI activity was normalized to leaf protein content. The protein content of the extracts used for the TPI assay was determined using a Bradford-assay (*Bradford, 1976*) on a 96-well microtiter plate.

## Quantification of total soluble proteins

Total soluble proteins were extracted from 50 mg of frozen ground leaf lamina (exact tissue masses were recorded) in 0.5 ml 0.1 M M Tris-HCl (pH 7.6) following the protocol described by *Ullmann-*

*Zeunert et al. (2012)*. Protein concentrations were measured using a Bradford assay (*Bradford, 1976*) on a 96-well microtiter plate using bovine serum albumin (BSA) as standard.

## Quantification of free amino acids

Free amino acids were extracted from leaf lamina by acidified MeOH extraction [MeOH:$H_2O$: HCOOH 15:4:1 (v/v/v)] and analyzed by liquid chromatography coupled to a triple quadrupole MS (Bruker EVOQ Elite, Bruker Daltonics, Bremen, Germany; www.bruker.com), as previously described (*Schäfer et al., 2016*).

## Measurement of starch, glucose, fructose and sucrose with a hexokinase assay

Glucose, fructose, sucrose and starch were determined following the protocol described by Machado and colleagues (*Machado et al., 2015*). Briefly, 100 mg plant tissues (exact tissue masses were recorded) were extracted first with 80% (v/v) ethanol and later twice with 50% (v/v) ethanol, each by incubation for 20 min at 80°C. Supernatants from all extractions were pooled, and sucrose, glucose and fructose were quantified enzymatically as previously described (*Velterop and Vos, 2001*). The remaining pellets were used for an enzymatic determination of starch content (*Smith and Zeeman, 2006*).

## Photosynthetic measurements

Net $CO_2$ assimilation rate was measured with a LI-COR LI-6400/XT portable photosynthesis system (LI-COR Inc., Lincoln, NE, US). All measurements were conducted using a 2 cm$^2$ chamber, at constant $CO_2$ (400 μmol $CO_2$ mol air$^{-1}$), light (300 μmol m$^{-2}$ s$^{-1}$ PAR), temperature (25–26°C) and relative humidity (20–40%). Measurements of photosynthetic rates of leaves with clip-cage were specifically done on the area included in the clip-cage. We measured photosynthetic rates of control leaves and leaves damaged by *T. notatus*. Leaves with clip-cages were analyzed in the covered area shortly after removal of the clip cage.

## Chlorophyll measurement

Chlorophyll was quantified using a Minolta SPAD Chlorophyll meter 502. Chlorophyll content is displayed in arbitrary SPAD units. Each sample value is the mean of chlorophyll content measured at three different random spots from each analyzed leaf. Leaves with clip cages were analyzed in the covered area shortly after the removal of the clip cage.

## qPCR quantification of cytokinin-related transcripts

RNA was extracted with TRIzol (Thermo Fisher Scientific, Waltham, MA, US), according to the manufacturer's instructions. cDNA was synthesized by reverse transcription using oligo(dT) primer and RevertAid reverse transcriptase (Thermo Fisher Scientific). Real-time qPCR was performed using actin as a standard on a Mx3005P qPCR machine (Stratagene, San Diego, CA, US) using a qPCR Core kit for SYBR Green I No ROX mix (Eurogentec, Seraing, Belgium). The primer sequences are provided in *Supplementary file 2*.

## Measurement of phytohormones

Levels of defense signaling compounds after three days of *T. notatus* herbivory were quantified as described by *Kallenbach et al., 2010*). JA, JA-Ile, OPDA and SA were analyzed as described by Kallenbach and colleagues (*Kallenbach et al., 2010*) and ABA as described by Dinh and colleagues (*Đinh et al., 2013*).

Kinetics of CKs, JA and JA-Ile during 144 hr of *T. notatus* attack was measured as previously described (*Schäfer et al., 2016*). In brief, phytohormones were extracted from ca. 100 mg of fresh ground leaf material (exact sample masses were recorded) using acidified methanol and purified on reversed phase and cation exchange solid-phase extraction columns. The measurements were done via liquid chromatography coupled to a triple quadrupole MS (Bruker EVO-Q Elite) equipped with a heated electrospray ionization source. The method was extended for the detection of $^{15}N$-labeled CKs. The parent → product ion transitions for $^{15}N$ labeled CKs are listed in *Supplementary file 3*. Chromatograms of IP, [D$_6$]-IP, [$^{15}N_5$]-IP as well as IPR, [D$_6$]-IPR and [$^{15}N_5$]-IPR are shown in *Figure 4—*

*figure supplement 4* and *5*. The same extraction method was used for CK extraction from *T. notatus*, using approximately 10 mg of ground material (five pooled samples of ca. 20 adults; exact sample masses were recorded).

## Cytokinin transfer experiment

*N. attenuata* plants with more than 98% of their total nitrogen content as $^{15}$N were obtained following the protocol described by Ullmann-Zeunert and colleagues (*Ullmann-Zeunert et al., 2012*). Briefly, twelve days after germination, plants were transferred into individual 50 mL hydroponic culture containers with Ca($^{15}$NO$_3$)$_2$ as the sole nitrogen source. Ten days later, the plants were transferred to 1 L hydroponic culture chambers with the same $^{15}$NO$_3$- concentration as of K$^{15}$NO$_3$. Once per week, the plants were fertilized with 1 mM K$^{15}$NO$_3$ and the containers were maintained at 1 L with distilled water.

To generate $^{15}$N labeled *T. notatus*, we reared insects for an entire generation on fully $^{15}$N-labelled *N. attenuata* plants. Two-hundred adult females were transferred to a 47.5 × 47.5 × 93 cm insect cage with four $^{15}$N-labelled *N. attenuata* plants in the early elongation stage of growth. Females were allowed to lay eggs for four days and subsequently removed. $^{15}$N-labelled plants were fertilized once a week (as described above), and after three weeks two fresh $^{15}$N-labelled plants were added to the insect cage. One week after the first adults emerged, the $^{15}$N labeled mirids were collected and used for the cytokinin transfer experiment.

Two types of experiments were conducted to quantify the transfer of cytokinins from mirids to *N. attenuata* plants. In the first, twenty $^{15}$N-labelled *T. notatus* adults were quickly anesthetized with CO$_2$ prior to clip-caging on a lower stem leaf of a $^{14}$N-grown plant, with one clip-cage per plant. We collected and froze in liquid nitrogen the leaf lamina corresponding to the area included in the clip-cage and thus damaged by mirids at different time-points: 0, 3, 6, 24, 48, 72, 96 and 120 hr. Each harvest time included at least three replicate plants. Samples were stored at −80°C until analysis.

In the second type of experiment, five $^{15}$N-labelled *N. attenuata* plants were placed in the mirid rearing cage. One lower stem or rosette leaf per plant was harvested at 0, 3, 6, 24, 48, 72, 96 and 120 hr, thus each harvest time represented five replicate plants. Plants were transferred once per day to a different cage to ensure that mirids did not accumulate $^{15}$N-labeled metabolites. We separated the leaf lamina from the mid-rib and froze the lamina in liquid nitrogen, and stored the samples at −80°C until analysis.

## Damage distribution of WT plants under field and glasshouse conditions

In the field, the damaged area on different leaf types was estimated as % of the total leaf area. The proportion of damaged leaf area for each leaf was visually estimated and the leaves were grouped into three different types (*Figure 6—figure supplement 1a*). Finally, we calculated average leaf damage for all leaf types: rosette leaves, the first (oldest) three stem leaves and all younger stem leaves and side branches. Similarly, *T. notatus* damage distribution within the plant was evaluated under controlled conditions in the glasshouse. In this experiment, a total of seven WT plants were used to form four replicates. The replicates consisted of three pairs of WT plants and one single plant, where each replicate was placed in one cage, meaning that one cage represented one replicate with no matter if there were one or two plants inside the cage. The plants in each cage were exposed to adults of *T. notatus* (n = 10 insects/plant) for one week. *T. notatus* infestations continued for an additional two weeks, where in the last week, five insects/plant were added to each cage. *T. notatus* damage was estimated from high resolution pictures of 15 leaves per plant at standardized rosette, mid stem, and young stem positions. Using Photoshop (Adobe), the damage was evaluated and expressed as a percentage of total damage per plant.

## Choice assays

For choice assays conducted in the field between young and fully mature leaves, we collected insects from native populations at our field station in Utah, USA. Ten to fifteen *T. notatus* adults were placed in a plastic cup. The cup was connected to two other plastic cups (*Figure 6—figure supplement 1b*), one enclosing a fully mature stem leaf and the other enclosing young, growing leaves (apical meristem and young leaves which had not yet completed the sink-source transition). To

prevent desiccation, leaf petioles were submerged in water in a 2 mL plastic microcentrifuge tube. As the insects are night-active, after one night (12 hr) during which the insects could choose between the two containers, we counted the number of mirids in each.

For choice assays between WT and ir*chk2/3* plants, we placed plants in a large mesh-enclosed cage in the glasshouse (3 × 4 × 1.6 m) into which 500 *T. notatus* were released. The damage on each plant (as described above) was estimated 10 days later.

Data from choice assays on i-ov*ipt* plants were taken from a previously published dataset (*Schäfer et al., 2013*). Plants were either treated with pure lanolin (LAN) as a control or with DEX-containing lanolin as described above. We treated the first (oldest) ten stem leaves of a flowering plant and placed one DEX- and one LAN-treated plant in one 47.5 × 47.5×93 cm insect cage. About 100 *T. notatus* adults were added to the cage, and the damaged leaf area was estimated after 10 days. The average damage level from all 10 treated leaves from each plant was counted as one replicate.

## Statistical analyses

Data were analyzed using R 3.3.1 (2016-06-21; http://www.r-project.org). Statistical tests and number of replicates, as well as transformations to data in order to meet assumptions of a test (homoscedasticity, normality), are provided in the figure legends. Normality of data sets was assessed by Shapiro–Wilk tests and homoscedasticity by Levene's test. If not mentioned otherwise, time course data were analyzed with ANCOVA with mirid feeding as factor and time as continuous explanatory variable. If the response variable was not linearly dependent on time we used two-way ANOVAs (TWA) with mirid feeding and time as factors. In all the analyses of experiments with the data from clip-cages, we only used data from control clip-cages and clip-cages with mirids. Differences were considered significant when $p < 0.05$.

## Acknowledgements

This work was funded by the Max-Planck-Society. In addition, Christoph Brütting, Stephan Meldau and Meredith C Schuman were funded by ERC Advanced Grant no. 293926 from the European Research Council to Ian T Baldwin. Martin Schäfer and Cristina Crava were funded by Collaborative Research Centre 'Chemical Mediators in Complex Biosystems - ChemBioSys' (SFB 1127) from the DFG. We thank Brigham Young University for the use of their Lytle Preserve field station, Mario Kallenbach, Matthias Schöttner, Thomas Hahn, Antje Wissgott, Wibke Kröber, Celia Diezel, and Eva Rothe for technical assistance, Claire Poore, Thomas Steier, Anja Hartmann, Spencer Arnesen and Katrina Welker for help with sample processing, Tamara Krügel, Andreas Weber, Andreas Schünzel and the entire glasshouse team for plant cultivation, and Rayko Halitschke for helpful discussions.

## Additional information

### Competing interests

Ian T Baldwin: Senior editor, *eLife*. The other authors declare that no competing interests exist.

### Funding

| Funder | Grant reference number | Author |
| --- | --- | --- |
| Max-Planck-Gesellschaft | Open-access funding | Christoph Brütting<br>Cristina Maria Crava<br>Martin Schäfer<br>Meredith C Schuman<br>Stefan Meldau<br>Nora Adam<br>Ian T Baldwin |
| European Commission | ERC Advanced Grant no. 293926 | Christoph Brütting<br>Meredith C Schuman<br>Stefan Meldau |

| Deutsche Forschungsge-meinschaft | Collaborative Research Centre SFB 1127 | Cristina Maria Crava Martin Schäfer |

The funders had no role in study design, data collection and interpretation, or the decision to submit the work for publication.

### Author contributions
Christoph Brütting, Conceptualization, Data curation, Formal analysis, Investigation, Visualization, Methodology, Writing—original draft, Writing—review and editing; Cristina Maria Crava, Conceptualization, Data curation, Formal analysis, Investigation, Methodology, Writing—original draft, Writing—review and editing; Martin Schäfer, Conceptualization, Investigation, Writing—review and editing; Meredith C Schuman, Conceptualization, Supervision, Writing—review and editing; Stefan Meldau, Conceptualization, Project administration; Nora Adam, Formal analysis, Investigation; Ian T Baldwin, Conceptualization, Supervision, Funding acquisition, Project administration, Writing—review and editing

### Author ORCIDs
Christoph Brütting https://orcid.org/0000-0001-9698-4400
Cristina Maria Crava https://orcid.org/0000-0003-3774-4567
Meredith C Schuman http://orcid.org/0000-0003-3159-3534
Ian T Baldwin http://orcid.org/0000-0001-5371-2974

### Decision letter and Author response
Decision letter https://doi.org/10.7554/eLife.36268.032
Author response https://doi.org/10.7554/eLife.36268.033

## Additional files

### Supplementary files
• Supplementary file 1. Calculations of the minimum amount of IP transferred by a single mirid in clip-cage experiment and estimation of the number of feeding mirids required to transfer the measured amount of IP in the whole-plant experiment.
DOI: https://doi.org/10.7554/eLife.36268.025

• Supplementary file 2. Sequences of primers used for real-time qPCR.
DOI: https://doi.org/10.7554/eLife.36268.026

• Supplementary file 3. Multi-reaction monitoring settings for the quantification of $[^{14}N_5]$-, $[^{15}N_5]$- and deuterated cytokinins in positive ionization mode.
DOI: https://doi.org/10.7554/eLife.36268.027

• Transparent reporting form
DOI: https://doi.org/10.7554/eLife.36268.028

### Data availability
All data generated or analysed during this study are available on Dryad Digital Repository

The following dataset was generated:

| Author(s) | Year | Dataset title | Dataset URL | Database, license, and accessibility information |
| --- | --- | --- | --- | --- |
| Brütting C, Crava CM, Schäfer M, Schuman MC, Meldau S, Adam N, Baldwin IT | 2018 | Data from: Cytokinin transfer by a free-living mirid to Nicotiana attenuata recapitulates a strategy of endophytic insects | http://dx.doi.org/10.5061/dryad.hq48044 | Available at Dryad Digital Repository under a CC0 Public Domain Dedication |

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
