## [Decision Letter]

Thank you for submitting your article "Cytokinin transfer by a free-living mirid to *Nicotiana attenuata* recapitulates a strategy of endophytic insects" for consideration by *eLife*.

Your article has been reviewed by an external peer reviewer, and the evaluation has been overseen by Joerg Bohlmann as the Reviewing Editor and Detlef Weigel as the Senior Editor. The external reviewer, Enric Frago, agreed to reveal his identity. We discussed the review and the Reviewing Editor has drafted this decision to help you prepare a revised submission.

Overall, the external review was positive. Based on this assessment and based on the Reviewing Editor's reading of the paper and our discussion, we believe the work presented in this manuscript will be of considerable interest to the broad audience of *eLife*. This paper has strong potential to impact how people view plant herbivore interactions. There are, however, a few issues that will require revisions. First, there are questions about the origin of the cytokinins. Preferentially, you will provide additional data that clearly document the insect origin of cytokinins. In their absence, please be more circumspect in your discussion of cytokinin origin. Additional work to test the plant fitness consequences is needed to substantially improve the paper.

External Reviewer's Comments

In this study Brütting, Crava and co-authors study how an herbivorous mirid bug manipulates the physiology of its host plant using plant hormone cytokinins (CKs). These hormones are important regulators of source-sink nutrient balances in plants, so that a potential benefit of increasing CKs in the local area the insect feeds on is an increase in the nutritional value of that area. This strategy has already been suggested as important in endophytic insects because their larvae are spatially constrained and cannot look for areas with an optimal nutritional value (particularly after plant defences are triggered). Several studies have shown that CK levels are higher in bodies of endophytic insects as well as in plant areas in which they feed, but direct CK production by insects remains elusive. This manuscript is thus very exciting because it shows for the first time (to my knowledge) that a CK, which is quite likely produced by the insect is incorporated into the plant, leading to transcriptomic changes and to associated nutritional changes. In addition, this effect is shown in a non-endophytic insect, which suggests that the mechanism reported here can be widespread in insects. The manuscript is well written and will be of interest to a large audience. The experiments are well designed because they follow a sequence of discoveries and novel hypotheses, which are subsequently tested. I have, however, few concerns regarding the interpretation of results that I detail below. I am not an expert on CK metabolism, so I will have minor comments in this regard.

One of the main concerns I have is that in the Results and Discussion sections authors speculate too much when interpreting their results. I have the feeling that the authors try to build a nice story, and they often interpret their results in a way that they point too directly to the following experiment.

For example, an important conclusion of this study is that thanks to CK manipulation, mirids can feed on their host plants without altering the nutritional quality of the plant. It is true that several metabolites like sugars were not altered, but proteins were, and the authors still state that "mirid feeding did not dramatically affect the overall nutritional quality of leaves" (subsection “*T. notatus* feeding does not negatively affect the nutritional quality of the attacked leaves”). It seems that authors are just picking the results they are interested in to keep the flow of the story. To what extent are protein contents important for this insect species, for instance? Most phloem feeders, for example, are strongly affected by protein contents in phloem, while sugars are in fact meaningless in terms of their consequences for insect fitness.

Related to my previous comment, the authors speculate about the benefits insects can gain from CK manipulation (e.g. Discussion section) but no formal test is performed. This study reports a suite of really complicated experiments, and I find very surprising that the authors did not perform such a simple experiment. Testing how insects perform on the two mutant plants that constitutively express or do not express CKs would be a quite simple and highly informative experiment. Alternatively, can they explain why this experiment is not needed to support their conclusions?

Also related to my first comment, I am still a bit puzzled by the enigmatic effect of mirids on plant fitness (e.g. subsection “Altered CK metabolism in *N. attenuata* affects its interaction with *T. notatus*”, Discussion section). As far as I know, CKs can alter nutrient balances among different parts of the plant, but how is it possible that an insect takes biomass from the plant without imposing any fitness cost? I am aware of the studies published by some of the authors of the present study in which overcompensation is suggested as a plant strategy to minimise damage when herbivory occurs. However, to my taste, the authors rely too much on these previous studies to argue that plants do not suffer from these mirids. In fact, the authors found in one of their experiments that plants indeed suffer when large insect densities are present. I think the authors should cling a bit less on their previous results and argue that maybe, plants can resist a certain level of herbivory, but they suffer when a certain threshold is reached. Why not test plant fitness consequences when insects feed on mutant plants that constitutively express CKs? This would unequivocally prove the costs of being able of triggering defences or not. If not possible, please de-emphasise discussions on the null effect of mirid feeding on plant fitness.

I find the arguments that CKs originate from the insect a bit unconvincing. The authors argue that CKs come from the insect because (i) CK concentration in the insect was ten times larger than in the plant (subsection “*T. notatus* contains high levels of IP” and Discussion section), and because (ii) these CK levels remained stable after five days of rearing on artificial diet. However, insects were reared on natural plants, then fed 5 days artificial diet, and then CKs measured. To my view, this experiment does not really demonstrate that CKs are synthesised by the insect as it is possible they are sequestered from the plant (as happens in other insects that sequester defensive plant metabolites). I wonder what would happen with CK levels if insects were reared since early instars on artificial diet.

---

## [Author Response]

External Reviewer's CommentsIn this study Brütting, Crava and co-authors study how an herbivorous mirid bug manipulates the physiology of its host plant using plant hormone cytokinins (CKs). These hormones are important regulators of source-sink nutrient balances in plants, so that a potential benefit of increasing CKs in the local area the insect feeds on is an increase in the nutritional value of that area. This strategy has already been suggested as important in endophytic insects because their larvae are spatially constrained and cannot look for areas with an optimal nutritional value (particularly after plant defences are triggered). Several studies have shown that CK levels are higher in bodies of endophytic insects as well as in plant areas in which they feed, but direct CK production by insects remains elusive. This manuscript is thus very exciting because it shows for the first time (to my knowledge) that a CK, which is quite likely produced by the insect is incorporated into the plant, leading to transcriptomic changes and to associated nutritional changes. In addition, this effect is shown in a non-endophytic insect, which suggests that the mechanism reported here can be widespread in insects. The manuscript is well written and will be of interest to a large audience. The experiments are well designed because they follow a sequence of discoveries and novel hypotheses, which are subsequently tested. I have, however, few concerns regarding the interpretation of results that I detail below. I am not an expert on CK metabolism, so I will have minor comments in this regard.One of the main concerns I have is that in the Results and Discussion sections authors speculate too much when interpreting their results. I have the feeling that the authors try to build a nice story, and they often interpret their results in a way that they point too directly to the following experiment.

Thanks for this insightful point. We agree that in parts we may have forced transitions and in reworking the text we have tried to find a more natural, less forced, flow for the experimental narrative and have reworked the explanatory text that we thought was helping readers follow our experimental logic. Throughout this work, we have strived to consider alternative hypotheses at each step in the process; and in parts, we may have overdone it, and confounded the logical flow with a heterogeneous consideration of the mountain of data that was collected in the process. We hope that this revision has been successful regarding this important point.

For example, an important conclusion of this study is that thanks to CK manipulation, mirids can feed on their host plants without altering the nutritional quality of the plant. It is true that several metabolites like sugars were not altered, but proteins were, and the authors still state that "mirid feeding did not dramatically affect the overall nutritional quality of leaves" (subsection “T. notatus feeding does not negatively affect the nutritional quality of the attacked leaves”). It seems that authors are just picking the results they are interested in to keep the flow of the story. To what extent are protein contents important for this insect species, for instance? Most phloem feeders, for example, are strongly affected by protein contents in phloem, while sugars are in fact meaningless in terms of their consequences for insect fitness.

We agree with the reviewer’s point. We do not know how and if protein content is important for *T. notatus* which unlike aphids, is not an exclusive phloem feeder, but we recognize that nitrogen is often a constraint for the diets of many herbivorous insects. Our results revealed that the decrease in total soluble proteins (TSPs) was only significant when we exposed the entire plant directly to our mirid colony and not when we confined twenty mirids to a single leaf (even when, in this latter case, feeding damage was clearly apparent). This was surprising because wounding and application of *M. sexta* oral secretions significantly decreases total soluble proteins in elicited *N. attenuata* leaves. We have revised the text in the Results session as follows to temper our conclusions:

“In summary, when only twenty mirids were allowed to feed on a single leaf the overall nutritional quality was not altered, although the feeding damage was visibly severe. In contrast, during a more extreme mirid infestation in which entire plants were severely attacked, TSP levels of attacked leaves decreased, but sugar and starch contents remained unchanged”.

Related to my previous comment, the authors speculate about the benefits insects can gain from CK manipulation (e.g. Discussion section) but no formal test is performed. This study reports a suite of really complicated experiments, and I find very surprising that the authors did not perform such a simple experiment. Testing how insects perform on the two mutant plants that constitutively express or do not express CKs would be a quite simple and highly informative experiment. Alternatively, can they explain why this experiment is not needed to support their conclusions?

We agree completely with the reviewer that, at least on paper, this experiment should provide the cleanest test of our hypothesis. However, constitutive CK over-expression or silencing is well known to result in such severe developmental abnormalities that the analysis of the effects of CKs on herbivore performance would be confounded by the accompanying morphological and developmental consequences of the CK manipulations. These vast pleiotropic effects of CKs are likely the reason why the plant-herbivore community has been slow to realize the importance of CKs in mediating responses to attack. The work presented in this paper was motivated by an observation published in previous paper (Schäfer et al., 2013), in which we developed a DEX-inducible system for *N. attenuata*, and as a proof of concept, used it to over-express a CK biosynthetic gene (IPT). When we planted these plants into the field and elicited particular leaves with DEX, the treated leaves were heavily attacked by mirids. This was the first direct evidence that mirids preferred high CK levels. In this paper, we use the same DEX-inducible system to over-express IPT and a transgenic line silenced in two of three CK receptors to manipulate CK levels in fully developed leaves to partially disentangle the pleiotropy. By quantifying damage levels, we show that mirids prefer leaves that over-express CKs compared to WT leaves, and that they prefer WT leaves to leaves impaired in CK perception. These data point to the importance of CKs in the mirid performance or at least feeding behavior.

Also related to my first comment, I am still a bit puzzled by the enigmatic effect of mirids on plant fitness (e.g. subsection “Altered CK metabolism in N. attenuata affects its interaction with T. notatus”, Discussion section). As far as I know, CKs can alter nutrient balances among different parts of the plant, but how is it possible that an insect takes biomass from the plant without imposing any fitness cost? I am aware of the studies published by some of the authors of the present study in which overcompensation is suggested as a plant strategy to minimise damage when herbivory occurs. However, to my taste, the authors rely too much on these previous studies to argue that plants do not suffer from these mirids. In fact, the authors found in one of their experiments that plants indeed suffer when large insect densities are present. I think the authors should cling a bit less on their previous results and argue that maybe, plants can resist a certain level of herbivory, but they suffer when a certain threshold is reached. Why not test plant fitness consequences when insects feed on mutant plants that constitutively express CKs? This would unequivocally prove the costs of being able of triggering defences or not. If not possible, please de-emphasise discussions on the null effect of mirid feeding on plant fitness.

We agree with the reviewer that in one of the two experimental designs used in this study, when we exposed entire *N. attenuata* plants to our glasshouse mirid colony, plants were heavily attacked and clearly suffered. We chose this heavy infestation setup because attack from large number of mirids was advantageous to clearly trace the transfer of labeled CKs from mirids into the CK pools of plants. As we had used this experimental setup for the labeling experiments, we were compelled to use the same setup with the same high herbivore load to examine effects on nutrient contents. We ran a parallel lower infestation setup and found no significant differences between the two setups for all nutrients examined except for TSPs which were significantly depleted under the heavy infestation condition. However, in our three decades of field work with *N. attenuata*, such high mirid densities do not commonly occur (in most years, plants in the field host between 3 to 20 individuals and not hundreds as was the case for these glasshouse experiments). Hence drawing conclusions about the fitness consequences from unrealistically high heavy infestation rates does not seem appropriate to us. Therefore, we rely more on the previous published data from our group (Kessler et al., 2004) because this work was conducted under field conditions at our field site in Utah. We think that these were the best possible circumstances in which to quantify plant fitness cost. Fitness data from field experiments should always hold sway over those from glasshouse experiments, which are frequently confounded by all sorts of artifacts (lack of AMF interactions, pot-bound roots, no UVB fluence, to name a few). Moreover, published experiments with the same mirid species attacking *Datura wrightii* conducted by other researchers reported similarly neutral fitness effects of *T. notatus* attack (see Elle and Hare, 2000; Hare and Elle, 2002).

We agree with reviewer that a formal analysis of the fitness effects of mirid attack on *N. attenuata* is beyond the scope of this paper and hence we have deleted discussions of the neutral effects of mirid feeding on plant fitness.

As noted above, while the proposed experiments using mutant plants that constitutively express CKs sound good on paper, they would not be illuminating due to the strong development effects of over-expressing CKs.

I find the arguments that CKs originate from the insect a bit unconvincing. The authors argue that CKs come from the insect because (i) CK concentration in the insect was ten times larger than in the plant (subsection “T. notatus contains high levels of IP” and Discussion section), and because (ii) these CK levels remained stable after five days of rearing on artificial diet. However, insects were reared on natural plants, then fed 5 days artificial diet, and then CKs measured. To my view, this experiment does not really demonstrate that CKs are synthesised by the insect as it is possible they are sequestered from the plant (as happens in other insects that sequester defensive plant metabolites). I wonder what would happen with CK levels if insects were reared since early instars on artificial diet.

We agree with the reviewer that the experiments have not rigorously ruled out the possibility that CKs found in mirids could have originated from host plant and were subsequently sequestered by the insect. Unfortunately, we not been successful (despite many attempts) in rearing *T. notatus* entirely on artificial diet and as a consequence, we have not been able to rigorously test this hypothesis. We have removed the text where we speculated about the origins of the CKs transferred by *T. notatus* in both the Results and Discussion sessions.